# REIMAGINING FEDERATED KNOWLEDGE GRAPH EMBEDDING WITH ONE-SHOT ADAPTIVE SEMANTIC ALIGNMENT

## ABSTRACT

Knowledge graphs (KGs) are widely used in various knowledge-driven applications, such as the recent surge in large language models. Yet, its adoption across different organizations is often hindered by data silos and privacy concerns, resulting in fragmented knowledge and limited performance. Federated learning (FL) has recently emerged as a potential solution for KG sharing while preserving data privacy. However, most federated KG embedding methods incur high communication and computation costs due to iterative updates and heavy synchronization, yet yield only limited performance gains, with additional privacy leakage risk via multiple communications. To address these challenges, we propose **OFKGE**, an efficient **O**ne-Shot **F**ederated **K**nowledge **G**raph **E**mbedding framework. Clients independently train local KGE models and upload them once, eliminating iterative communication. The server distills and aggregates semantic patterns from submitted models to form a generalized global representation via the Adaptive Semantic Alignment (ASA), which is then redistributed as optimized auxiliary knowledge. Clients locally fine-tune their models using this distilled global guide, enhancing inference performance while preserving personalized structural information. OFKGE achieves competitive representation quality with minimal communication, making it suitable for resource-constrained and privacy-sensitive federated settings. Experimental results on the MKG-W, MKG-Y, DB15K, FB15K-237, and NELL-995 datasets demonstrate that OFKGE significantly reduces resource overhead while outperforming state-of-the-art methods on most metrics. The code is available at https://anonymous.4open.science/r/OFKGE-CACB.

## 1 INTRODUCTION

*Knowledge graphs* (KGs) encode real-world entities and their semantic relations, and are widely used in tasks like question answering (Huang et al., 2019; Wang et al., 2024b), personalized recommendation (Wang et al., 2019; Yang et al., 2022), and large language model (LLM) reasoning (Wang et al., 2024a; Tan et al., 2025). However, completing KGs across different organizations is often hindered by data silos and privacy constraints (Chen et al., 2024). To address this challenge, *Federated Learning* (McMahan et al., 2017; Yang et al., 2019) (FL) has emerged as a promising solution for collaborative KG embedding without exposing raw data.

Despite these advances, existing federated knowledge graph embedding (FKGE) methods (Chen et al., 2021; 2022; Zhu et al., 2023; Zhang et al., 2025) suffer from high communication overhead, modest performance gains, and increased privacy risks. For instance, FedE (Chen et al., 2021) transmits about 1.35 billion parameters over multiple rounds, while FedLU (Zhu et al., 2023) nearly doubles training time with similar communication. Besides costs, multi-round exchanges also increase vulnerability to privacy attacks (Hu et al., 2023). In contrast, prior studies have shown that *one-shot federated learning* (OFL) (Guha et al., 2019; Zhang et al., 2022a; Bai et al., 2025), which aggregates each client's model only once, substantially reduces communication costs and inherently enhances privacy (see Appendix A.4 for detailed analysis of one-shot federated knowledge graph).

A more fundamental limitation lies in embedding aggregation. Most FKGE methods rely on static weighted aggregation on the server side, failing to capture the semantic diversity among different

clients. This results in coarse or distorted global embeddings that inadequately represent client knowledge, thereby constraining performance improvements. For example, FedE, FedLU, and NP-FedKGC (Liu et al., 2025) adopt simple average aggregation, while methods like PFedEG (Zhang et al., 2025) and GFedKG (Wang et al., 2024c) incorporate client relations or neighbor information to compute aggregation weights. However, these strategies still lack dynamic adaptability to varying client semantics.

To tackle the above issues, we explore the OFL paradigm for knowledge graph embedding. While OFL has been previously explored in domains such as computer vision (Zhang et al., 2022a; Diao et al., 2023; Heinbaugh et al., 2023), its potential in FKGE remains underexplored. Motivated by this gap, we propose OFKGE, an efficient One-Shot Federated Knowledge Graph Embedding framework enhanced by an Adaptive Semantic Alignment (ASA) mechanism. At its core, ASA enables the server to dynamically aggregate semantic knowledge by selecting the most informative teacher model from client embeddings based on the current training context. On the client side, we introduce a regularization-based fine-tuning strategy to effectively integrate global guidance while preserving local semantics. This design yields more effective and personalized embeddings for downstream tasks while significantly reducing communication overhead and privacy risks.

Our contributions are summarized as follows: ❶ We propose OFKGE, the first OFL framework tailored for knowledge graph embedding. By requiring only a single round of communication, OFKGE greatly reduces communication overhead and privacy risks, making it well-suited for scenarios involving high-sensitivity data and limited resources. ❷ We introduced an adaptive knowledge integration strategy, which features an Adaptive Semantic Alignment mechanism on the server. This mechanism selects the most informative client model for global distillation, enabling refined alignment between global and local semantics. Meanwhile, on the client side, regularization-based fine-tuning is applied to balance global guidance and local knowledge, enabling performance improvement while preserving local personalization. ❸ Extensive experiments on multiple datasets demonstrate that OFKGE outperforms state-of-the-art FKGE methods with substantially lower communication and computation costs.

## 2 RELATED WORKS

OFL addresses the communication and computation bottlenecks of traditional multi-round FL by building a global model in a single communication round, typically using average-based, ensemble-based, or knowledge distillation methods (Guha et al., 2019; Zhou et al., 2020; Tang et al., 2024). Average-based methods (Qu et al., 2022; Jhunjhunwala et al., 2023; Yadav et al., 2023) improve model averaging by incorporating additional information to better weight local models. Ensemble approaches, like FedOV (Diao et al., 2023), aggregate client predictions, while distillation methods, such as DENSE (Zhang et al., 2022a) and FedCAVE (Heinbaugh et al., 2023), train a student model using auxiliary or synthetic data. Advanced variants like Co-Boosting (Dai et al., 2024) and IntactOFL (Zeng et al., 2024) enhance performance via dynamic weighting or self-supervised generation. Most OFL methods focus on image tasks, whereas we are the first to explore OFL for KGs. In FKGE, prior work typically performs entity aggregation (Chen et al., 2021; Wang et al., 2024c; Zhang et al., 2025), relation aggregation (Zhang et al., 2022b), or joint aggregation (Hu et al., 2022), often using FedAvg (Chen et al., 2021; 2022; Zhu et al., 2023) or GNN-based (Wang et al., 2024c; Liu et al., 2025) schemes. While effective for privacy-preserving knowledge sharing, these approaches still face limitations from communication costs and privacy risks due to repeated transmissions. The detailed related works are listed in Appendix A.3.

## 3 PRELIMINARY

Assume a set of clients $\{C_1, C_2, \ldots, C_C\}$, where each client holds a local knowledge graph $G_c = (E_c, R_c, T_c)$, forming the federated set $\mathcal{G}^{fed} = \{G_c \mid 1 \leq c \leq C\}$. Entities or relations may overlap across clients (e.g., $e \in E_i \cap E_j$ for $i \neq j$), but due to privacy constraints, clients cannot share raw data. Furthermore, in the **One-Shot Federated Setting**, communication is limited to a single round of client-to-server interaction.

Under this setting, each client uploads its model parameters only once. The server then integrates models $\{M_c\}$ to construct a global embedding $(E_g, R_g)$ as auxiliary knowledge $K$, optimizing:

$$f_{global} = \min_{(E_g, R_g)} \mathcal{L}(E_g, R_g; M_1, M_2, \ldots, M_C). \tag{1}$$

Each client $c$ subsequently learns personalized embeddings using its local triples $T_c$ and global knowledge $K_c$ from the server, optimizing:

$$f_{local}^c = \min_{(E_c, R_c)} \mathcal{L}(E_c, R_c; G_c, K_c). \tag{2}$$

While privacy is not the main focus, mechanisms like differential privacy (Lin et al., 2020) can be integrated. The server is assumed to maintain private entity/relation alignment tables (Chen et al., 2021; Zhang et al., 2022b), and may use unlabeled or synthetic data for distillation (Lin et al., 2020). The detailed preliminary is listed in Appendix A.2.

## 4 METHODOLOGY

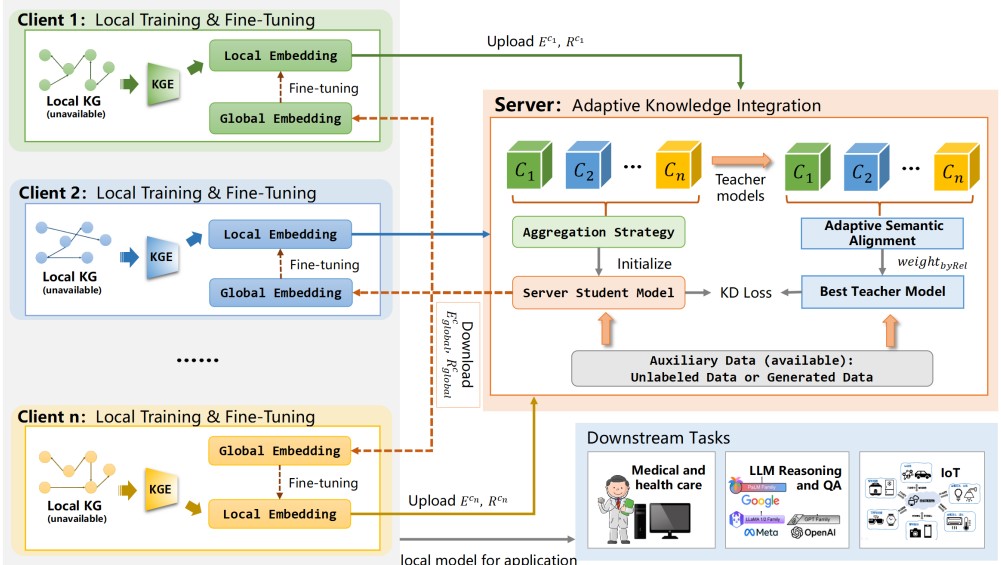

Figure 1: Overall Framework of OFKGE. The process consists of three stages: **One-Shot Model Distillation**, **Server-Side Adaptive Knowledge Integration**, and **Client-Side Local Fine-Tuning**. This can be likened to the paradigm of *knowledge for payment* in reality: clients provide their contributions (local models) as *payment* to the knowledge provider (server). The server then integrates knowledge from various clients to form a richer, more comprehensive global understanding, which benefits everyone involved.

Under the OFL setting, OFKGE follows a three-stage training process, as illustrated in Figure 1. Specifically, each client uploads its local model to the server once to receive global knowledge, which enhances the local models and reduces communication and training costs. The server integrates this knowledge using the Adaptive Knowledge Integration Module, where client models serve as teachers and the global model as the student. The best teacher is selected for each epoch to guide the training of the global model. In the final stage, the server provides tailored global knowledge to each client, who refines its model using local data and a regularization-based strategy. This process enables clients to incorporate global knowledge while maintaining the uniqueness of their local data.

### 4.1 ONE-SHOT MODEL DISTILLATION

The following section, together with Algorithm 1, will provide a detailed description of the one-shot communication process in clients under the OFKGE framework.

---

**Algorithm 1:** Workflow of OFKGE

---

**Input:** Number of clients $C$; Local training rounds $T_{local}$; Gobal training rounds $T_{global}$;
Client-local KG data $\mathcal{G}_c$; Unlabeled or generated data $D$ for distillation
**Output:** Fine-tuned client model $M_{c,1}$

---

1 **Server executes:**
2 Initialize $E_0$, $R_0$ and distribute to clients;
3 Receive client models and weights: $\{M_c, w_c\}$;
4 Initialize global model $(E^0_{\text{global}}, R^0_{\text{global}})$ according to $\{M_c\}$ using *FedAvg*;
5 Sample data $D$ and split into mini-batches $\mathcal{B}$;
6 **for** $t \leftarrow 1$ **to** $T_{global}$ **do**
7    **foreach** *batch* $b \in \mathcal{B}$ **do**
8       $M_{\text{teach}} \leftarrow \text{ASA}(b, \{M_c\}, \{w_c\})$, Eq. 4;
9       Compute loss $\mathcal{L}_{\text{global}}$, Eq. 6;
10       Update $(E^t_{\text{global}}, R^t_{\text{global}})$;
11 Project and distribute global knowledge $(E^c_{\text{global}}, R^c_{\text{global}})$ to clients;

12 **Each client $c$ executes:**
13 Receive $(E^c_0, R^c_0)$ from server;
14 **Phase 1: Local Training**;
15 $(E^c_0, R^c_0) \leftarrow \text{LOCALTRAINING}(E^c_0, R^c_0)$, detailed in Algorithm 2;
16 Construct local model: $M_{c,0} \leftarrow (E^c_0, R^c_0)$;
17 Evaluate relation-aware weights: $w_c$;
18 Upload $(M_{c,0}, w_c)$ to server;
19 **Phase 2: Fine-tuning**;
20 Receive $(E^c_{\text{global}}, R^c_{\text{global}})$ from server;
21 $(E^c_1, R^c_1) \leftarrow \text{FINETUNING}(M_{c,0}, E^c_{\text{global}}, R^c_{\text{global}})$, detailed in Algorithm 4;
22 Update local model for downstream tasks $M_{c,1} \leftarrow (E^c_1, R^c_1)$;

---

The server first initializes global entity and relation embeddings $(E_0, R_0)$ for all clients. To prevent privacy leakage from direct sharing, these embeddings are transformed via client-specific permutation matrices $P^c_{ent}$ and $P^c_{rel}$, so that client $c$ receives $E^c_0 \leftarrow P^c_{ent} E_0$ and $R^c_0 \leftarrow P^c_{rel} R_0$.

Client $c$ then initializes its local model with these embeddings and trains it using a classical knowledge graph embedding method such as TransE (see Appendix A.5.1 for algorithm details). The model employs the scoring function $f(h, r, t) = -\|h + r - t\|$. Its loss function is defined as:

$$\mathcal{L}_{kge} = \sum_{(h,r,t)_{pos}} \sum_{(h',r,t')_{neg}} [\gamma + f(h, r, t) - f(h', r, t')]_+ \tag{3}$$

where $h, r, t$ denote the embeddings of the head entity, relation, and tail entity respectively; $(h, r, t)_{pos}$ and $(h', r, t')_{neg}$ are the sets of positive and negative samples; and $\gamma$ is the margin hyperparameter. After training, the client obtains its local model $M_{c,0} \leftarrow (E^c_0, R^c_0)$.

To assist the server in selecting the optimal teacher model among clients, the client evaluates its performance across all relation types using training and validation sets. Metrics such as Hits@K ($K \in \{1, 3, 10\}$) and MMR are computed to construct a relation-aware performance matrix $weight^c_{byRel} = w^\top \cdot metrics \in \mathbb{R}^{|\mathcal{R}|}$, reflecting the model's capability across different relations. where $w$ is a vector of weights and *metrics* is a vector of selected evaluation indicators.

Finally, in the one-shot communication phase, each client uploads the trained model parameters $M_{c,0}$ and the relation-aware performance matrix $weight^c_{byRel}$.

## 4.2 SERVER-SIDE ADAPTIVE KNOWLEDGE INTEGRATION

To enable adaptive knowledge integration, the server first collects the models $\{M_{c,0}\}^C_{c=1}$ uploaded by clients as candidate teachers. It then aggregates these models using a parameter-weighted strategy such as *FedAvg* (see Appendix A.5.3 for alternative strategies) to initialize the global embeddings $(E^0_{global}, R^0_{global}) \leftarrow \text{FedAvg}(M_{1,0}, \ldots, M_{c,0})$, which serve as the starting point for the student model on the server side (see Appendix A.5.2 for algorithm details of server-side adaptive knowledge integration).

To avoid the use of sensitive labeled data, OFKGE adopts privacy-friendly unlabeled data or samples synthesized by pretrained generators as training data for knowledge distillation. Specifically, the server constructs the distillation dataset $\mathcal{D}$ by sampling from a predefined open unlabeled dataset or using a pretrained data generator (see Appendix A.5.5 for details on the data synthesis procedure). This approach not only protects the privacy of the original data but also provides more diverse information, helping the model to absorb global knowledge and improve generalization. During each distillation iteration, for every batch $b$, the server dynamically selects the most suitable client

model $M_{teach}$ as the teacher model based on the relation-aware performance matrix $weight_{byRel}^c$ uploaded by the clients. The adaptive semantic alignment can be defined as:

$$M_{teach} = \arg\max_{M_c} \sum_{r \in \mathcal{R}(b)} weight_{byRel}^c(r) \tag{4}$$

where $\mathcal{R}(b)$ denotes the set of relations involved in the current batch $b$. This mechanism enables the server to select the most semantically appropriate teacher for each group of samples, achieving fine-grained knowledge transfer. The server then uses the selected teacher model to compute its score on the sample and generates the soft label output $f_{tea}^k$, which is compared with the score $f_{stu}^k$ of the current student model. This soft label is used to guide the student model's update, and the smoothed L1 loss is computed as follows:

$$\mathcal{L}_{distill-soft} = \begin{cases} \frac{1}{2}(f_{tea}^k - f_{stu}^k)^2, & |f_{tea}^k - f_{stu}^k| \leq 1 \\ |f_{tea}^k - f_{stu}^k| - \frac{1}{2}, & otherwise \end{cases} \tag{5}$$

where $k$ denotes the $k$-th sample, and $f_{tea}^k, f_{stu}^k$ are the scores of the teacher and student models for this sample, respectively. In addition, the server uses the loss function from the original knowledge graph embedding model (e.g., TransE) as the hard label loss $\mathcal{L}_{kge}$. Here, $\alpha \in [0,1]$ controls the balance between the hard and soft labels during global training. The final composite loss function is then defined as:

$$\mathcal{L}_{distillation} = \alpha\mathcal{L}_{kge} + (1-\alpha)\mathcal{L}_{distill-soft} \tag{6}$$

The server updates the global model parameters according to the composite loss. Through multiple training epochs, the server fuses multi-source knowledge into the global embedding $E_{global}, R_{global}$. These embeddings are then projected and distributed to each client as their global auxiliary knowledge $K^c \leftarrow (E_{global}^c, R_{global}^c)$, where $E_{global}^c \leftarrow P_{ent}^c E_{global}$ and $R_{global}^c \leftarrow P_{rel}^c R_{global}$.

## 4.3 CLIENT-SIDE LOCAL FINE-TUNING

After initial local training and uploading its model, the client receives globally distilled entity and relation embeddings from the server. These serve as global auxiliary knowledge to further guide the fine-tuning of the local model. The goal of this phase is to utilize global knowledge to optimize the local embeddings, enhance the generalization capability of the model, and compensate for the information limitations caused by training on local data alone. Ultimately, this enables the construction of more complete and semantically rich embeddings, thereby providing stronger support for downstream tasks.

Taking client $c$ as an example, it performs training based on its local knowledge graph data $\mathcal{G}_c$, while incorporating the global auxiliary knowledge $\mathcal{K}^c$ returned by the server (see Appendix A.5.4 for algorithm details). The training objective for this phase is defined as:

$$\mathcal{L}_{\text{fine-tuning}} = \mathcal{L}_{\text{kge}} + \beta\mathcal{L}_{re}(E_1^c; E_0^c, \mathcal{K}^c) \tag{7}$$

where $\beta \in [0,1]$ is a hyperparameter that balances local learning and global guidance. $\mathcal{L}_{\text{kge}}$ denotes the knowledge graph embedding loss function (e.g., TransE) used to model triples, and $\mathcal{L}(E_1^c; E_0^c, \mathcal{K}^c)$ is a regularization term introduced to guide the update direction of the local model using global knowledge. This term ensures that the current embeddings do not deviate significantly from the original local model, while maximizing the integration of global auxiliary knowledge. The regularization term is defined as:

$$\mathcal{L}(E_1^c; E_0^c, \mathcal{K}^c) = \log\left[1 + \frac{\exp\left(\text{sim}(E_1^c, E_0^c)\right)}{\exp\left(\text{sim}(E_1^c, E_{\mathcal{K}}^c)\right)}\right] \tag{8}$$

where $\text{sim}(\cdot, \cdot)$ denotes a similarity function, $E_1^c$ represents the current local entity embeddings during fine-tuning, $E_0^c$ is the entity embeddings uploaded to the server from the previous round,

and $E_\mathcal{K}^c$ denotes the global auxiliary entity embeddings provided by the server. This design ensures that the local model can retain personalized characteristics while effectively learning from global knowledge, thereby achieving further completion and generalization of the knowledge structure.

# 5 EXPERIMENTS

## 5.1 DATASETS

Experiments are conducted on five public datasets, including three multimodal knowledge graph datasets (MKG-W, MKG-Y, DB15K) and two standard knowledge graph datasets (FB15K-237, NELL-995). These datasets vary in scale, covering different numbers of entities, relations, and triples, as summarized in Appendix A.6.1. This diversity allows us to assess the adaptability of the proposed model across different data scenarios. To construct the federated setting, each dataset is randomly partitioned into multiple local subsets (clients) with partial overlap in entities and relations. For each client, we split the data into training, validation, and test sets in an 8:1:1 ratio.

## 5.2 BASELINES

To validate the effectiveness of OFKGE, we compare it against several state-of-the-art FKGE models under the one-shot federated setting, including: FedE (Chen et al., 2021), FedR (Zhang et al., 2022b), FedM (Hu et al., 2022), FedLU (Zhu et al., 2023), and PFedEG (Zhang et al., 2025). Additionally, we include two baseline strategies: Single and Ensemble. Single means each client trains independently using its own data but is aware of the full set of entities and relations across all clients. Ensemble means a model ensemble approach that mimics averaged-based OFL by weighting and combining the independently trained local models. We exclude baselines such as FedProx (Li et al., 2020) and FedEC (Chen et al., 2022) since they require at least two communication rounds per aggregation, which contradicts the one-shot constraint. Further details are provided in Appendix A.6.3.

## 5.3 IMPLEMENTATION

OFKGE is implemented based on OpenKE (Han et al., 2018) and evaluated on a Linux server (Ubuntu 24.04.2) equipped with an NVIDIA GeForce RTX 3070 Ti GPU. We adopt TransE (Bordes et al., 2013) for representation learning with 200-dimensional embeddings (for validation experiments with RotatE, see Appendix A.6.5), optimized using Adam (Kinga et al., 2015). Unless otherwise specified, the number of clients is fixed at 3. On the server side, knowledge distillation is performed for 1000 epochs with a trade-off parameter $\alpha = 0.4$ and a learning rate of 0.001. Client-side fine-tuning includes a regularization term weighted by $\beta = 0.8$. All baselines are reproduced from their official open-source implementations and adapted to our data splits; unrelated components (e.g., the unlearning module in FedLU) are removed to maintain consistency with the one-shot setting. We evaluate link prediction performance using Mean Reciprocal Rank (MRR) and Hits@N (Appendix A.6.2), with additional implementation details provided in Appendix A.6.4.

## 5.4 RESULTS

To evaluate the effectiveness of the proposed OFKGE method, we conducted experiments on five representative knowledge graph datasets. As shown in Table 1, OFKGE consistently outperforms all baselines, frequently ranking first or second. This demonstrates its strong generalization and its ability to capture global semantic information to enhance client performance.

OFKGE outperforms Single, Ensemble, and existing FKGE methods across all datasets and clients, demonstrating its effectiveness in integrating heterogeneous knowledge under one-shot communication. This improvement stems from server-side distillation and client-side regularization, which capture global semantic correlations. Compared to PFedEG, OFKGE achieves more consistent performance due to fine-grained relation-level semantic alignment, enabling more effective knowledge integration. Minor fluctuations on datasets like MKG-Y highlight challenges with imbalanced data, suggesting that our relation-specific strategies could further enhance stability. Detailed case study is provided in Appendix A.6.10.

Table 1: Performance Comparison

| Dataset | Model | Client 1 | | | | Client 2 | | | | Client 3 | | | |
|---|---|---|---|---|---|---|---|---|---|---|---|---|---|
| | | MRR | Hit@10 | Hit@3 | Hit@1 | MRR | Hit@10 | Hit@3 | Hit@1 | MRR | Hit@10 | Hit@3 | Hit@1 |
| MKG-W | Single | 17.22 | 32.10 | 23.11 | 8.37 | 16.32 | 31.43 | 22.05 | 7.50 | 15.57 | 30.62 | 21.46 | 6.80 |
| | Ensemble | 18.90 | 34.68 | 24.39 | 9.90 | 17.84 | 34.27 | 23.35 | 8.47 | 17.15 | 33.63 | 23.08 | 7.70 |
| | FedE | 20.74 | 39.14 | 27.81 | 10.25 | 20.38 | 38.62 | 27.20 | 9.78 | 19.60 | 37.82 | 26.41 | 8.86 |
| | FedR | 17.34 | 32.78 | 22.61 | 8.70 | 16.77 | 32.21 | 22.69 | 7.93 | 15.94 | 31.22 | 21.73 | 7.22 |
| | FedM | 20.98 | 38.46 | 28.14 | 10.42 | 20.41 | 38.47 | 26.97 | 9.93 | 19.33 | 37.76 | 26.41 | 8.57 |
| | FedLU | 20.86 | 38.40 | 28.18 | 10.44 | 20.32 | 38.58 | 26.56 | 9.97 | 19.38 | 37.86 | 26.22 | 8.55 |
| | PFedEG | 20.92 | 38.58 | 28.12 | 10.42 | 20.39 | 38.99 | 27.16 | 9.85 | 19.57 | 37.86 | 26.49 | 8.86 |
| | **OFKGE** | **21.81** | **40.61** | **29.34** | **10.52** | **21.50** | **40.62** | **29.18** | **9.97** | **20.59** | **39.81** | **28.29** | **9.02** |
| MKG-Y | Single | 33.42 | 36.14 | 34.95 | 31.36 | 28.17 | 30.62 | 29.58 | **26.18** | 12.59 | 14.95 | 13.71 | 10.94 |
| | Ensemble | 33.68 | 36.77 | 35.16 | 31.43 | 28.16 | 30.69 | 29.58 | 25.97 | 12.56 | 14.54 | 13.22 | 10.94 |
| | FedE | 34.00 | 37.48 | 35.37 | 31.64 | 27.65 | 31.31 | 29.37 | 25.13 | 13.12 | 15.85 | 13.91 | 11.21 |
| | FedR | 33.58 | 37.20 | 35.30 | 31.22 | 27.92 | 30.62 | 29.30 | 25.83 | 13.03 | 15.78 | 13.85 | 11.21 |
| | FedM | 34.35 | 37.62 | 35.79 | 32.06 | 27.64 | 31.38 | 28.88 | 25.20 | 12.82 | 15.99 | 13.78 | 10.59 |
| | FedLU | 33.65 | 37.06 | 34.95 | 31.43 | 28.01 | 31.38 | 29.09 | 25.97 | 13.00 | 15.92 | 14.12 | 10.94 |
| | PFedEG | 34.09 | 37.55 | 35.51 | 31.71 | 28.25 | 31.52 | 29.93 | 25.90 | 13.02 | 15.72 | 13.78 | 11.08 |
| | **OFKGE** | **34.74** | **38.11** | **36.00** | **32.48** | **28.65** | **32.22** | **30.20** | 26.11 | **13.45** | **16.41** | **14.47** | **11.42** |
| DB15K | Single | 22.69 | 44.78 | 30.97 | 9.87 | 23.40 | 45.29 | 32.14 | 10.42 | 22.60 | 44.63 | 31.29 | 9.56 |
| | Ensemble | 23.26 | 45.73 | 32.23 | 9.94 | 23.61 | 46.39 | 32.55 | 10.23 | 23.27 | 45.60 | 31.97 | 10.05 |
| | FedE | 24.03 | 47.63 | 32.83 | 10.42 | 24.10 | 47.63 | 33.20 | 10.31 | 24.09 | 47.43 | 33.52 | 10.30 |
| | FedR | 23.26 | 45.38 | 31.86 | 10.36 | 23.71 | 46.90 | 32.27 | 10.69 | 23.45 | 45.34 | 32.43 | 10.47 |
| | FedM | 23.91 | 47.34 | 32.91 | 10.27 | 24.40 | 47.97 | 33.70 | 10.56 | 24.10 | 47.30 | 33.44 | 10.29 |
| | FedLU | 23.70 | 47.27 | 32.62 | 10.10 | 24.56 | 47.96 | 33.70 | 10.87 | 24.13 | 46.98 | 33.00 | 10.62 |
| | PFedEG | 23.75 | 47.16 | 32.43 | 10.22 | 24.53 | 47.55 | 32.99 | 11.25 | 24.05 | 47.03 | 33.42 | 10.32 |
| | **OFKGE** | **24.95** | **49.68** | **34.52** | **10.59** | **25.50** | **49.99** | **34.93** | **11.28** | **25.10** | **49.07** | **34.57** | **10.92** |
| FB15K -237 | Single | 30.42 | 55.93 | 37.51 | 17.08 | 30.82 | 56.47 | 37.87 | 17.46 | 30.40 | 55.57 | 37.41 | 17.16 |
| | Ensemble | 30.50 | 56.13 | 37.34 | 17.30 | 30.89 | 56.39 | 37.77 | 17.68 | 30.36 | 55.48 | 37.33 | 17.16 |
| | FedE | 30.27 | 55.44 | 37.47 | 17.04 | 30.52 | 55.53 | 37.42 | 17.41 | 30.38 | 55.41 | 37.50 | 17.20 |
| | FedR | 30.36 | 55.51 | 37.51 | 17.09 | 30.51 | 55.73 | 37.58 | 17.26 | 30.44 | 55.60 | 37.42 | 17.23 |
| | FedM | 30.33 | 55.47 | 37.19 | 17.20 | 30.44 | 55.63 | 37.60 | 17.21 | 30.46 | 55.43 | 37.44 | 17.37 |
| | FedLU | 30.45 | 55.47 | 37.36 | 17.31 | 30.57 | 55.57 | 37.47 | 17.45 | 30.40 | 55.47 | 37.36 | 17.28 |
| | PFedEG | 30.59 | 56.14 | 37.49 | 17.34 | 31.07 | 56.49 | 38.04 | 17.89 | 30.70 | 55.77 | 37.61 | 17.61 |
| | **OFKGE** | **31.03** | **57.05** | **38.17** | **17.55** | **31.39** | **57.18** | **38.66** | **17.95** | **31.09** | **56.58** | **38.14** | **17.75** |
| NELL -995 | Single | 18.56 | 35.62 | 23.36 | 9.20 | 17.78 | 35.48 | 22.63 | 8.17 | 18.22 | 35.55 | 23.21 | 8.62 |
| | Ensemble | 19.44 | 38.23 | 24.77 | 9.13 | 19.06 | 39.02 | 24.83 | 8.22 | 19.33 | 38.43 | 25.24 | 8.56 |
| | FedE | 21.07 | 40.75 | 27.16 | 10.15 | 20.53 | 41.13 | 26.71 | 9.16 | 20.53 | 40.14 | 26.82 | 9.34 |
| | FedR | 19.17 | 37.38 | 24.09 | 9.37 | 18.75 | 37.95 | 24.15 | 8.38 | 18.96 | 37.38 | 24.41 | 8.92 |
| | FedM | 21.07 | 40.68 | 27.24 | 10.08 | 20.61 | 40.85 | 27.10 | 9.20 | 20.81 | 40.51 | 27.47 | 9.63 |
| | FedLU | 20.66 | 40.18 | 26.91 | 9.62 | 20.44 | 41.03 | 26.54 | 9.08 | 20.64 | 40.54 | 26.91 | 9.45 |
| | PFedEG | 20.96 | 40.94 | 27.20 | 9.90 | 20.65 | 41.01 | 26.80 | **9.33** | 20.38 | 40.08 | 26.99 | 9.12 |
| | **OFKGE** | **22.07** | **42.90** | **29.05** | **10.22** | **21.47** | **43.12** | **28.38** | 9.30 | **22.11** | **43.13** | **29.74** | **9.74** |

## 5.5 ABLATION STUDY

Table 2: Ablation Study on MKG-W, DB15K, and FB15K-237. *AI* denotes the server aggregation initialization; *ASA* denotes Adaptive Semantic Alignment; *FT* denotes client local fine-tuning.

| Dataset | Model | Client 1 | | | | Client 2 | | | | Client 3 | | | |
|---|---|---|---|---|---|---|---|---|---|---|---|---|---|
| | | MRR | Hit@10 | Hit@3 | Hit@1 | MRR | Hit@10 | Hit@3 | Hit@1 | MRR | Hit@10 | Hit@3 | Hit@1 |
| MKG-W | w/o AI | 20.57 | 37.68 | 27.44 | 10.02 | 20.03 | 38.37 | 27.49 | 9.17 | 18.91 | 36.90 | 26.26 | 8.20 |
| | w/o ASA | 20.41 | 38.25 | 28.04 | 9.34 | 20.28 | 38.19 | 27.74 | 9.21 | 18.76 | 37.19 | 26.47 | 7.59 |
| | w/o FT | 16.38 | 31.63 | 22.36 | 7.40 | 17.68 | 33.03 | 23.51 | 8.69 | 14.56 | 29.72 | 20.21 | 5.82 |
| | **OFKGE** | **21.81** | **40.61** | **29.34** | **10.52** | **21.50** | **40.62** | **29.18** | **9.97** | **20.59** | **39.81** | **28.29** | **9.02** |
| DB15K | w/o AI | 24.13 | 47.87 | 33.32 | 10.19 | 24.36 | 47.86 | 33.73 | 10.53 | 24.14 | 47.60 | 33.62 | 10.36 |
| | w/o ASA | 24.10 | 47.90 | 33.41 | 10.14 | 24.75 | 48.32 | 33.92 | 10.92 | 24.17 | 48.12 | 33.80 | 10.05 |
| | w/o FT | 22.47 | 44.89 | 31.20 | 9.22 | 22.94 | 45.35 | 32.04 | 9.69 | 22.59 | 44.72 | 31.65 | 9.27 |
| | **OFKGE** | **24.95** | **49.68** | **34.52** | **10.59** | **25.50** | **49.99** | **34.93** | **11.28** | **25.10** | **49.07** | **34.57** | **10.92** |
| FB15K -237 | w/o AI | 30.62 | 56.15 | 37.55 | 17.40 | 30.87 | 56.31 | 37.77 | 17.70 | 30.50 | 55.67 | 37.53 | 17.29 |
| | w/o ASA | 30.73 | 56.34 | 37.56 | 17.50 | 31.06 | 56.65 | 38.00 | 17.81 | 30.62 | 55.81 | 37.55 | 17.42 |
| | w/o FT | 29.91 | 55.49 | 36.74 | 16.71 | 30.31 | 55.64 | 37.16 | 17.15 | 29.99 | 55.20 | 36.97 | 16.81 |
| | **OFKGE** | **31.03** | **57.05** | **38.17** | **17.55** | **31.39** | **57.18** | **38.66** | **17.95** | **31.09** | **56.58** | **38.14** | **17.75** |

As shown in Table 2, the full OFKGE model achieves the best performance across all datasets, confirming the importance of each component. Removing any module leads to a performance drop. The largest decline occurs without Fine-Tuning (*w/o FT*), highlighting its key role in preserving

client-specific features. Removing Aggregated Initialization (*w/o AI*) also causes a notable drop, as AI ensures consistent initialization and convergence. Excluding Adaptive Semantic Alignment (*w/o ASA*) has a smaller but still negative impact, showing its value in reducing semantic discrepancies across clients. Ablation studies on the regularization term (Eq. 8) used in the FT stage can be found in Appendix A.6.6.

## 5.6 COST ANALYSIS

Table 3: Training Time and Communication Parameter Size on MKG-W, DB15K, and FB15K-237

| Model | MKG-W | | DB15K | | FB15K-237 | |
|---|---|---|---|---|---|---|
| | Training Time(s) | Parameters(M) | Training Time(s) | Parameters(M) | Training Time(s) | Parameters(M) |
| FedE | 836.00 | 1354.64 | 1238.50 | 1507.14 | 3177.98 | 5579.60 |
| FedR | 833.00 | 24.24 | 1235.84 | 39.41 | 3176.36 | 80.10 |
| FedM | 839.70 | 1371.98 | 1241.86 | 1538.82 | 3190.74 | 5603.12 |
| FedLU | 1858.52 | 1300.18 | 2015.03 | 992.20 | 6313.04 | 4565.16 |
| **OFKGE** | **358.58** | **20.68** | **508.02** | **23.19** | **1056.81** | **26.35** |

Table 3 compares the training time and communication cost of OFKGE and baselines across three datasets. OFKGE achieves the lowest training time and communication overhead, thanks to its one-shot client upload and server-side distillation. In contrast, baselines like FedE, FedM, and FedLU require multiple communication rounds, leading to higher costs, often exceeding 1000M parameters. Even FedR, which transfers only relation embeddings, incurs more overhead than OFKGE. Since OFKGE transmits once, its reported cost reflects the total, highlighting its clear advantage in communication-constrained settings.

## 5.7 VARIANTS

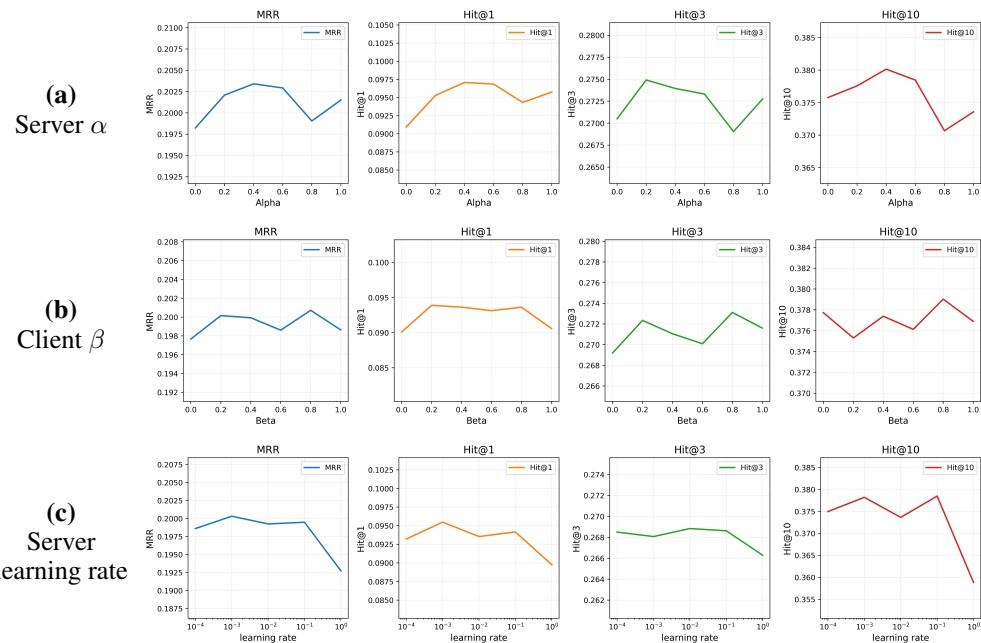

**(a)** Server $\alpha$

**(b)** Client $\beta$

**(c)** Server learning rate

Figure 2: Average Test Results of MKG-W under Different Hyperparameters

We first evaluate the scalability of OFKGE by varying the number of clients. As shown in Figure 3, all evaluation metrics consistently improve with more clients, unlike baselines such as FedE and FedLU, which often degrade. This is due to OFKGE's domain-specific partitioning strategy that allows overlapping entities and relations, better reflecting real-world scenarios. Such overlap enables the model to integrate complementary knowledge across clients, resulting in a more complete and

robust global representation. Additionally, we conducted experiments with low-overlap KGs (see Appendix A.6.7), demonstrating that our framework remains effective even under such conditions.

We further study the sensitivity of two key hyperparameters: the server-side distillation loss weight ($\alpha$) and the client-side regularization weight ($\beta$). As shown in Figures 2a and 2b, optimal performance is achieved around $\alpha = 0.4$ and $\beta = 0.8$. Setting $\alpha = 0$ or $\alpha = 1$ leads to performance drops, indicating the importance of combining hard-label supervision and teacher guidance. Similarly, omitting regularization ($\beta = 0$) weakens global alignment, while overly large $\beta$ values constrain local adaptation. Finally, we evaluate the server learning rate. Figure 2c shows that a learning rate of $10^{-3}$ provides the best performance. Larger rates ($10^{-2}$ or higher) cause

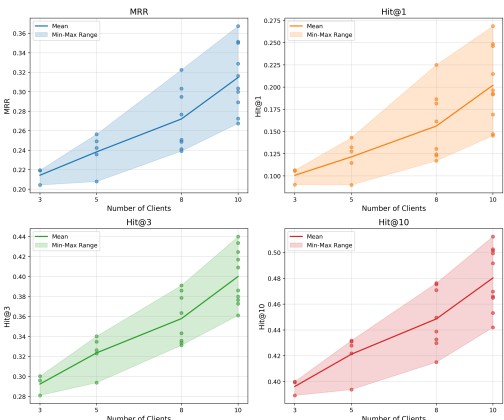

Figure 3: Average Results of MKG-W with Varying Numbers of Clients

instability, while smaller ones ($10^{-4}$) lead to underfitting, confirming that moderate server updates are key to effective knowledge distillation.

### 5.8 VISUALIZATION OF SYNTHETIC DATA

To compare synthetic data with real data, we present a selection of synthetic triples from the MKG-W dataset in Figure 4. While these synthetic triples reflect certain relational patterns of the real KG, they are not semantically equivalent to the original triples and may even contain factual inaccuracies. Despite this semantic gap and the presence of some factual errors, training with synthetic triples still enhances the distillation process, as demonstrated in Table 1. See Appendix A.6.8 for details.

| Real Triples | Synthetic Triples |
|---|---|
| *Triple*: (QuickSpot, developer, Namco) 
 *Rule*: (video game, developer, animation studio) | *Triple*: (Grid Seeker: Project Storm Hammer, publisher, Taito) 
 *Rule*: (video game, publisher, video game publisher) |
| *Triple*: (Namco, location of formation, Tokyo) 
 *Rule*: (animation studio, location of formation, metropolitan prefecture) | *Triple*: (TriStar Pictures, location of formation, Concord) 
 *Rule*: (film production company, location of formation, city in the United States) |
| *Triple*: (Bogach, located in/on physical feature, Outer Hebrides) 
 *Rule*: (village, located in/on physical feature, archipelago) | *Triple*: (Tall Maran, located in/on physical feature, Wellington) 
 *Rule*: (village, located in/on physical feature, urban area) |
| *Triple*: (Driven, original language of film or TV show, Spanish ) 
 *Rule*: (film, original language of film or TV show, natural language) | *Triple*: (The Madcap Laughs, original language of film or TV show, Romanian) 
 *Rule*: (album, original language of film or TV show, natural language) |

Figure 4: Visualization of Real and Synthetic Data

## 6 CONCLUSION

We propose OFKGE, a one-shot federated knowledge graph embedding framework that addresses the high communication, computation costs, and limited reasoning gains in existing FKGE methods. OFKGE streamlines training into three stages: one-shot client model upload, server-side knowledge distillation via a teacher-student approach, and client-side model fine-tuning. This design balances efficiency, personalization, and reasoning accuracy. Experiments on five datasets show that OFKGE achieves strong performance with significantly reduced overhead. Additionally, its distillation-based design supports model heterogeneity and is well-suited for future extension to multimodal knowledge graphs.

**Limitation** While OFKGE effectively reduces communication and computation costs, it has limitations. The one-shot upload mechanism may lack flexibility in dynamic or continuously evolving scenarios, and the effectiveness of server-side distillation depends on the quality of the uploaded client models. A detailed analysis of limitations is provided in Appendix A.7.

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

## A  APPENDIX

### A.1  THE USE OF LARGE LANGUAGE MODELS

The language of this paper was polished using large language models (LLMs) to enhance clarity and readability. The final content and academic integrity remain the responsibility of the authors.

### A.2  DETAILED PRELIMINARY

Assume a set of clients $\{C_1, C_2, \ldots, C_C\}$, where each client holds an independent local knowledge graph, defined as: $\mathcal{G}^{fed} = \{G_c = (E_c, R_c, T_c) \mid 1 \leq c \leq C\}$. Here, $E_c$, $R_c$, and $T_c$ denote the sets of entities, relations, and triples in client $c$, respectively. There may be partial overlap of entities or relations among different clients' knowledge graphs (e.g., $e \in E_i \cap E_j$ for $i \neq j$). However, due to privacy constraints, clients are not allowed to share any raw data, and communication is strictly limited to client-server interactions.

The core objective of a federated knowledge graph is to enable knowledge sharing across clients through multi-round interaction between clients and the server. Under this framework, each client $c$ aims to learn personalized entity and relation embeddings using its local triples $T_c$ and global auxiliary knowledge $K_c$ provided by the server. Let $\mathcal{L}$ denote the loss function defined on client $c$, the local optimization objective can be formulated as:

$$f_{local}^c = \min_{(E_c, R_c)} \mathcal{L}(E_c, R_c; G_c, K_c). \tag{9}$$

Unlike traditional multi-round federated learning, the **One-Shot Federated Setting** allows each client to upload its model parameters only once. Under this constraint, the server must extract and integrate knowledge from the uploaded client models $\{M_1, M_2, \ldots, M_C\}$ without relying on further communication, and construct a unified global embedding representation $(E_g, R_g)$, which will serve as the global auxiliary knowledge $K$ for future learning. The corresponding global optimization objective is:

$$f_{global} = \min_{(E_g, R_g)} \mathcal{L}(E_g, R_g; M_1, M_2, \ldots, M_C). \tag{10}$$

As privacy preservation is not the primary focus of this study, existing privacy-preserving mechanisms (e.g., differential privacy (Lin et al., 2020)) can be seamlessly integrated into the proposed method. To enable more effective model fusion, we make the following assumptions about the server: it privately maintains entity and relation alignment tables (Chen et al., 2021; Zhang et al., 2022b) to identify overlapping components across clients. Additionally, the server is allowed to utilize unlabeled data from other domains or generate synthetic data using pretrained generators (e.g., GAN (Goodfellow et al., 2014)) as distillation data (Lin et al., 2020).

## A.3 DETAILED RELATED WORKS

**One-shot federated learning**  Traditional federated learning relies on multi-round training to iteratively optimize models. However, in some scenarios, excessive communication and computation become major bottlenecks. OFL addresses this by building a global model through a single round of parameter transmission. While this may sacrifice some performance, it enhances privacy and significantly reduces communication overhead. Guha et al. (2019) first introduced OFL with two strategies: heuristic selection of a subset of client models for ensembling, and knowledge distillation using auxiliary data for aggregation. Current OFL methods largely fall into these two categories: ensemble-based and model distillation methods (Tang et al., 2024). Ensemble-based methods integrate client model predictions, as in Guha et al. (2019) ensemble mechanism. FedOV (Diao et al., 2023) further introduces open-set voting to tackle label inconsistency and improve generalization. Model distillation methods treat local models as teachers and use auxiliary data to train a student model on the server (Lin et al., 2020). For example, DOSFL (Zhou et al., 2020) transmits only distilled datasets, while DENSE (Zhang et al., 2022a) is the first data-free OFL approach, training a generator to support distillation. FedCVAE (Heinbaugh et al., 2023) reframes local training as learning conditional variational autoencoders, then distills the ensemble into a decoder to synthesize training data. Co-Boosting (Dai et al., 2024) combines adversarial sample generation with dynamic ensemble weighting for better performance. IntactOFL (Zeng et al., 2024) introduces a mixture-of-experts framework, preserving full local knowledge and generating samples via self-supervised learning that approximates the latent distribution of local data. Notably, existing OFL approaches almost exclusively focus on image-based tasks (Lin et al., 2020; Diao et al., 2023). To the best of our knowledge, we are the first to apply OFL to KGs. Unlike image-based tasks where alignment and fine-tuning are often unnecessary (Lin et al., 2020; Diao et al., 2023), knowledge graphs present unique structural and semantic challenges, requiring customized solutions. Our framework is specifically designed to overcome these challenges.

**Knowledge graph embedding**  Knowledge graph embedding (KGE) aims to represent entities and relations in a continuous low-dimensional vector space while preserving the structural and semantic information of the original graph. By converting complex graph-structured data into vector representations, KGE enables efficient computation and facilitates downstream tasks such as recommendation and reasoning. Existing KGE methods can be broadly categorized into translation-based, semantic matching-based, and neural network-based models. Translation-based approaches, inspired by the translational invariance observed in word embeddings, model relations as translation vectors that connect head and tail entities in the embedding space. A representative example is TransE (Bordes et al., 2013), which learns embeddings by minimizing the distance between the vector sum of the head entity and relation, and the tail entity (i.e., $\mathbf{h} + \mathbf{r} \approx \mathbf{t}$). Variants such as TransH (Wang et al., 2014) and RotatE (Sun et al., 2019) introduce relation-specific projections or dynamic mapping strategies to capture more complex relational structures. Semantic matching models, such as RESCAL (Nickel et al., 2011), DistMult (Yang et al., 2014), and ComplEx (Trouillon et al., 2016), use algebraic similarity functions, ranging from bilinear forms to complex-valued dot products, to score triples based on the compatibility of their embeddings. Meanwhile, neural network-based models employ deep architectures to capture intricate patterns in graph data. ConvE (Nguyen et al., 2018a) and ConvKB (Nguyen et al., 2018b) apply convolutional operations (Dettmers et al., 2018) over entity-relation pairs, while graph neural network models like R-GCN (Schlichtkrull et al., 2018) and CompGCN (Vashishth et al., 2020) incorporate neighborhood aggregation and relational composition to model structural dependencies. Although these methods have demonstrated strong performance in centralized settings, they face challenges in distributed or privacy-sensitive scenarios. FKGE seeks to overcome these limitations by enabling decentralized training without direct data sharing. Our work builds on these foundational embedding models, particularly translation-based approaches like TransE, and adapts them to a OFL framework to support efficient and privacy-preserving representation learning across clients.

**Federated knowledge graph**  Current research on federated knowledge graphs primarily aims to enable knowledge integration across distributed graphs via federated learning, allowing clients to benefit from diverse domains and perspectives. federated knowledge graphs has demonstrated value both in academic research and practical applications. In real-world scenarios, it supports tasks such as e-commerce recommendation and IoT service enhancement. For instance, FPKS (Sun et al., 2025) focuses on improving federated knowledge services in IoT environments, while

FedKGRec (Ma et al., 2024) targets personalized recommendations across domains such as e-commerce, video sharing, and online advertising. In academic research, federated knowledge graphs have been shown to facilitate knowledge completion and cross-domain learning without compromising data privacy. However, FKGE remains in its early stages, with only a few existing studies.

Currently, FKGE is easily confused with related research in Federated Graph Learning (FGL) and Federated Embedding Learning (FEL). Here, we provide a brief clarification of the differences. FGL extends federated learning to general graph structure data, where nodes contain feature vectors $x$ and corresponding labels $y$, enabling collaborative training among multiple clients via a trusted server without sharing raw graph data, and is commonly applied to tasks such as node classification (Wan et al., 2024; Yan et al., 2024). FEL, on the other hand, generalizes federated learning to the embedding space, where clients collaboratively learn vector representations of items or features, typically for recommendation or retrieval systems (Mao et al., 2025). FKGE differs from both FGL and FEL in that it focuses on encoding entities and relations within triples, aiming to capture structured relational knowledge rather than just node features or general embeddings.

Table 4: Comparison of FKGE Methods

| Methods | Federation Type | Communication | Transmission Type | Core Techniques |
|---|---|---|---|---|
| FedE | Centralized | Multi-round | Entity | Federated Averaging, Data Matching |
| FKGE | Decentralized | Multi-round | Entity, Relation | Differential Privacy, Adversarial Generation |
| FedR | Centralized | Multi-round | Relation | Global Sharing of Relation Embeddings |
| FedM | Centralized | Multi-round | Entity, Relation | Federated Averaging |
| FedEC | Centralized | Multi-round | Entity | Contrastive Learning |
| FedLU | Centralized | Multi-round | Entity | Mutual Knowledge Distillation, Unlearning |
| GFedKG | Centralized | Multi-round | Entity | GNN-based Aggregation |
| PFedEG | Centralized | Multi-round | Entity | Client-wise Relation Graph |
| OFKGE | Centralized | One-shot | Entity, Relation | Adaptive Semantic Alignment, Knowledge Distillation |

Existing FKGE methods can be classified into three categories based on the type of embeddings uploaded from clients to the server: Entity Aggregation, Relation Aggregation, and Joint Aggregation of Entity and Relation (as Table 4). In entity aggregation, clients transmit entity embeddings for aggregation, typically using the classic *FedAvg*, as seen in the FedE (Chen et al., 2021). Building on FedE, GFedKG (Wang et al., 2024c) introduces a GNN-based aggregation mechanism that incorporates neighboring features of entities from multiple clients to enhance the aggregation. PFedEG (Zhang et al., 2025), on the other hand, constructs a client-wise relation graph based on semantic relationships among clients and leverages it to perform personalized aggregation of entity embeddings. FedEC (Chen et al., 2022) extends FedE by introducing contrastive learning at the client side. NP-FedKGC (Liu et al., 2025) improves entity aggregation by explicitly modeling latent neighborhood information. FedLU (Zhu et al., 2023) further extends FedE by introducing mutual knowledge distillation between clients and server, and is the first to apply machine unlearning to FKGE tasks. Relation aggregation forms a global relation embedding at the server, as seen in FedR (Zhang et al., 2022b), which improves privacy and reduces communication costs by focusing on the relations of KG. Joint aggregation refers to separately aggregating both entity and relation embeddings, with FedM (Hu et al., 2022) being the first to jointly optimize both types of embeddings. In addition to these server-client architectures, there is also a peer-to-peer paradigm, where no centralized coordinator exists. In this decentralized setup, clients collaborate equally and directly exchange embedding updates. To the best of our knowledge, FKGE (Peng et al., 2021) is the only model that adopts this structure. These approaches demonstrate the feasibility of cross-domain and cross-organization knowledge sharing while preserving a certain level of data privacy. While these methods demonstrate the feasibility of knowledge sharing under privacy constraints, they still suffer from considerable communication overhead and heightened privacy risks due to frequent multi-round transmissions. For instance, on the FB15k-237-C3 dataset, FedE improves the mean reciprocal rank (MRR) by only 3.88% over local training, yet requires the transmission of approximately 1.35 billion parameters over multiple communication rounds (Chen et al., 2021). FedLU offers a modest further improvement of 3.21% in MRR over FedE, but nearly doubles the training time, with similar communication costs (Zhu et al., 2023). Worse still, these approaches expose a heightened privacy risk, as frequent multi-round transmissions significantly increase the potential for embedding leakage.

## A.4 ANALYSIS OF ONE-SHOT FEDERATED KNOWLEDGE GRAPH

### A.4.1 THEORETICAL ANALYSIS

OFL naturally enhances privacy and reduces communication overhead, which is particularly advantageous for one-shot federated knowledge graph embedding. Unlike traditional multi-round FKGE methods that iteratively exchange model embeddings, the one-shot approach performs a single aggregation of client embeddings on the server, lowering the risk of privacy leakage and minimizing communication costs.

**Effectiveness**    The core idea of OFKGE is to extract structured knowledge from each client's local KGE model via knowledge distillation and transfer it to the global student model on the server in a single-shot communication. The knowledge contained in a knowledge graph can be effectively transmitted through the distillation process (Liu et al., 2023; 2022; Zhu et al., 2023).

Specifically, assuming that the local KGs of all clients collectively form a global KG, each client can only observe a subset of the global KG, i.e., a local subgraph. Consequently, the client model captures the semantic and reasoning capabilities of this local subgraph, and the *knowledge* it encodes can be understood as the semantic patterns between entities and relations within the subgraph. On the server side, knowledge distillation allows these local *knowledge* fragments from all subgraphs to be integrated into a global student model. Therefore, extracting knowledge from client models via knowledge distillation and integrating it to form a global model is both reasonable and effective.

In our one-shot approach, the global model converges through knowledge distillation on the server side. Specifically, we first initialize the global model using an aggregation method such as FedAvg (other initialization strategies are detailed in Appendix A.5.3), and then gradually acquire knowledge from client local models by optimizing the knowledge distillation objective (Eqn 6). In each training step, operations such as gradient descent are applied to minimize this objective, allowing the student model to progressively approach the selected teacher models. In contrast, multi-round approaches rely on iterative client-server interactions to gradually achieve global convergence. Existing federated KGE methods (Chen et al., 2021; Zhu et al., 2023) mainly perform FedAvg on the server side (e.g., Eqn 13) and minimize the KGE objective (e.g., Eqn 3) on clients in each communication round. If multi-round methods are naively adapted to a one-shot setting, the server would only perform a single FedAvg, which is insufficient to fully transfer client knowledge.

Therefore, our one-shot method addresses the issue that simple aggregation in multi-round methods cannot fully capture the diverse knowledge from clients in a one-shot scenario. After aggregation, knowledge distillation is applied to effectively transfer client knowledge to the global model, enabling it to integrate more knowledge than traditional aggregation methods.

**Privacy**    OFL avoids iterative gradient exchanges, substantially mitigating privacy risks such as gradient inversion attacks. Prior studies (Hatamizadeh et al., 2023; Huang et al., 2021) have shown that intermediate gradients shared during multiple rounds can be exploited to reconstruct local training data. In contrast, one-shot aggregation collects client models or embeddings in a single step, greatly reducing the likelihood of reconstructing sensitive data from shared information.

Formally, let $P_{leak}(R)$ denote the probability of successful data reconstruction after $R$ rounds of communication, where $p$ is the per-round leakage probability. In multi-round FKGE:

$$P_{leak}(R) = 1 - (1-p)^R \qquad (11)$$

As $R \gg 1$, $P_{leak}(R) \to 1$, indicating that repeated embeddings exchanges make private data increasingly vulnerable. By contrast, in the one-shot setting, the leakage probability reduces to:

$$P_{leak}(1) = p \qquad (12)$$

Thus, one-shot FKGE inherently provides stronger protection against inversion and reconstruction attacks (Mendieta et al., 2025; Liu et al., 2024; Zeng et al., 2024).

### A.4.2 Experimental Analysis

Several studies have experimentally demonstrated that OFL provides advantages in both privacy protection and communication efficiency (Tang et al., 2024; Zhang et al., 2024). In the domain of FKGE, to the best of our knowledge, this work is the first to apply OFL to KGs. Therefore, we provide an experimental analysis from both privacy and communication efficiency perspectives.

**Privacy**  Existing studies have shown that attackers can successfully infer KG triples in victim clients via inference attacks (Hu et al., 2023). They proposed three inference attack methods, initiated from both server and client sides, which can successfully identify KG triples in victim clients, revealing the vulnerability of multi-round federated knowledge graphs in privacy protection. Notably, our framework prohibits the server-side generator from directly accessing real data and conducts only a single communication round, substantially reducing the risk of privacy leakage. Furthermore, in Section 5.8, we present the generated triples, which do not directly expose information from the real triples.

**Communication Efficiency**  As shown in Table 3, our framework exhibits significant advantages over multi-round federated learning in terms of communication overhead and model parameter transmission. Experimental results indicate that OFKGE achieves the lowest training time and communication cost compared to multi-round federated approaches.

### A.5 Algorithm description

In federated learning, single-round communication often leads to a loss of information integrity, thereby weakening the model's learning performance. This phenomenon is akin to a one-time conversation between people: due to hearing interference, vague expressions, or language barriers, information may be misunderstood, omitted, or even completely distorted. Through multi-round interactions, however, information can be progressively refined, clarified, and confirmed, leading to more accurate and comprehensive understanding.

Therefore, efficiently extracting and integrating valid knowledge from limited information remains a core challenge in OFL. To address this, **OFKGE** introduces a *One-shot Model Distillation* architecture, which aggregates models from multiple clients by distilling knowledge using unlabeled or synthetically generated data. This mechanism functions like processing, contrasting, and refining diverse perspectives (client models) using extensive prior knowledge (unlabeled or synthetic data), to identify the optimal information source (best teacher model), thereby obtaining more accurate and effective global knowledge.

---

**Algorithm 2:** Client-Side Initial Local Training

**Input:** Local training rounds $T_{\text{local}}$; local knowledge graph data $D_c$; score function $f$ of selected
      knowledge embedding model (e.g., TransE)
**Output:** Local model $M_{c,0}$; relation-aware weight $w_c$

1   Receive initial global embeddings from server: $E_0^c, R_0^c$;
2   Split local dataset $D_c$ into mini-batches: $\mathcal{B} = \{b_1, b_2, \ldots, b_n\}$;
3   **for** $t = 1$ *to* $T_{local}$ **do**
4      **foreach** *batch* $b \in \mathcal{B}$ **do**
5          Compute training loss: $\mathcal{L}_{\text{kge}} = \sum_{\text{pos}} \sum_{\text{neg}} [\gamma + f_{\text{pos}} - f_{\text{neg}}]$
6          Update embeddings: $E_0^c \leftarrow E_0^c - \eta \nabla \mathcal{L}_{\text{kge}}(b), R_0^c \leftarrow R_0^c - \eta \nabla \mathcal{L}_{\text{kge}}(b)$
7   Construct local model: $M_{c,0} \leftarrow (E_0^c, R_0^c)$;
8   Evaluate relation-aware weight: $w_c = w^\top \cdot metrics$
9   Upload $(M_{c,0}, w_c)$ to the server;

---

### A.5.1 Client Side: Initial Local Training

Algorithm 2 outlines the initial local training procedure on the client side within the OFKGE framework. Upon receiving the initial global entity and relation embeddings from the server, each client performs multiple rounds of training on its local knowledge graph data to optimize the embeddings. Additionally, it computes relation-aware weights to reflect the significance of local relations. The

resulting local model and weights are then uploaded to the server, providing a foundation for subsequent knowledge distillation and integration.

---

**Algorithm 3:** Server-Side Adaptive Knowledge Integration

---

**Input:** Number of clients $C$; global training rounds $T_{\text{global}}$; unlabeled or generated data $D$; entity mapping matrix $P_{ent}$; relation mapping matrix $P_{rel}$

**Output:** Final global embeddings $E_{\text{global}}, R_{\text{global}}$

---

1  Initialize global entity and relation embeddings $E_0, R_0$;
2  **foreach** *client* $c = 1$ *to* $C$ **do**
3      $E_0^c \leftarrow P_{ent}^c E_0$;
4      $R_0^c \leftarrow P_{rel}^c R_0$;
5      Distribute $E_0^c, R_0^c$ to client $c$;

6  Collect client models $\{M_{c,0}\}_{c=1}^{C}$ and their relation-aware weights $\{w_c\}_{c=1}^{C}$ as teacher models;
7  Initialize server student model: $(E_{\text{global}}^0, R_{\text{global}}^0) \leftarrow \text{FEDAVG}(\{M_{c,0}\})$;
8  Split unlabeled/generated data $D$ into mini-batches: $\mathcal{B} = \{b_1, b_2, \ldots, b_n\}$;
9  **for** $t = 1$ *to* $T_{global}$ **do**
10      **foreach** *batch* $b \in \mathcal{B}$ **do**
11          $M_{\text{teach}} \leftarrow \text{ASA}(\{M_{c,0}\}, \{w_c\})$;    // Adaptive Semantic Alignment (Eq. 4)
12          Compute global loss $\mathcal{L}_{\text{global}}$ using $M_{\text{teach}}$;    // Eq. 6
13          Update server student model:

$$E_{\text{global}}^t \leftarrow E_{\text{global}}^t - \eta \nabla \mathcal{L}_{\text{global}}(b), \quad R_{\text{global}}^t \leftarrow R_{\text{global}}^t - \eta \nabla \mathcal{L}_{\text{global}}(b)$$

14  **foreach** *client* $c = 1$ *to* $C$ **do**
15      $E_{\text{global}}^c \leftarrow P_{ent}^c E_{\text{global}}$;
16      $R_{\text{global}}^c \leftarrow P_{rel}^c R_{\text{global}}$;
17      Distribute $E_{\text{global}}^c, R_{\text{global}}^c$ to client $c$;

---

### A.5.2  SERVER SIDE: ADAPTIVE KNOWLEDGE INTEGRATION

In the **OFKGE** framework, the server is responsible for extracting global knowledge from the models uploaded by each client in a one-time transmission and constructing a unified global knowledge embedding representation. Specifically, the server uses a knowledge distillation mechanism to fuse local knowledge from multiple clients, generating shareable global entity and relation embeddings. These serve as global auxiliary knowledge to support subsequent model updates and downstream tasks on the client side.

Algorithm 3 outlines the server-side adaptive knowledge integration procedure in the OFKGE framework. The server first initializes global embeddings and distributes them to clients using pre-defined entity and relation mappings. After collecting the locally trained models and their corresponding relation-aware weights, the server aggregates them to form an initial global student model. During each global training round, the server performs knowledge distillation using unlabeled or generated data, guided by an adaptive semantic alignment strategy (ASA). The global embeddings are iteratively refined based on the distillation loss. Finally, the updated global embeddings are redistributed to each client to support subsequent model fine-tuning.

### A.5.3  INITIAL AGGREGATION STRATEGY

To effectively initialize the global embeddings $(E_{\text{global}}^0, R_{\text{global}}^0)$, the server aggregates the uploaded models $\{M_{c,0}\}$ from $C$ clients using one of the aggregation strategies described below. These strategies aim to balance the influence of local knowledge while addressing the heterogeneity among client-specific embeddings. However, this initialization may introduce noise due to domain-specific biases. To address this, the framework subsequently applies adaptive knowledge distillation (Appendix A.5.2) to correct and refine the initial global embeddings, enabling fine-grained knowledge transfer from clients. In the following, we present two representative methods and their extensions for global embedding initialization. While we primarily adopt FedAvg for our experiments, all the approaches described are compatible with our framework.

**Average-based Aggregation**   A commonly used strategy in FKGE is the average-based aggregation method, such as *FedAvg*. This approach performs a weighted average over entity and relation embeddings from all clients, where the weights are typically proportional to the size of each client's local dataset. In our context, a more targeted form of aggregation can be used (Chen et al., 2021; Zhu et al., 2023): only overlapping entities (or relations) that appear on multiple clients are aggregated. Specifically, if an entity $e$ appears in a subset of clients $\mathcal{C}_e$, its global embedding is calculated by averaging its embeddings from those clients:

$$\mathbf{E}_{\text{global}}(e) = \frac{1}{|\mathcal{C}_e|} \sum_{c \in \mathcal{C}_e} \mathbf{E}_c(e) \tag{13}$$

Entities or relations that are unique to a single client are retained as-is in the global model, serving as their initial global representations. Although this strategy is simple and efficient, it may neglect semantic and structural differences between clients, potentially leading to suboptimal performance under heterogeneous data distributions.

**Client Relation-based Aggregation**   To better capture structural heterogeneity among clients, a personalized aggregation mechanism based on inter-client relationships can be adopted. Inspired by PFedEG (Zhang et al., 2025), this method constructs a client similarity graph $\mathbf{A} \in \mathbb{R}^{C \times C}$, where each element $A_{ij}$ measures the similarity between client $i$ and client $j$. The similarity can be computed using several commonly adopted metrics based on shared entity embeddings:

- **Euclidean Distance**: Measures the absolute difference between entity embeddings in the vector space, capturing the overall proximity between client-specific models. This distance is sensitive to both magnitude and direction.

$$A_{ij} = \exp\left(-\|\mathbf{E}_i(e) - \mathbf{E}_j(e)\|_2\right) \tag{14}$$

- **Cosine Similarity**: Evaluates the angular similarity between embedding vectors, making it suitable for measuring the semantic alignment between clients, regardless of embedding magnitude.

$$A_{ij} = \frac{\mathbf{E}_i(e) \cdot \mathbf{E}_j(e)}{\|\mathbf{E}_i(e)\|_2 \cdot \|\mathbf{E}_j(e)\|_2} \tag{15}$$

- **Manhattan Distance**: Also known as $L_1$ distance, it sums the absolute differences across dimensions. Compared to Euclidean distance, it is more robust to noise and outliers, making it suitable for scenarios with unstable embeddings.

$$A_{ij} = \exp\left(-\sum_{k=1}^{d} \left|E_i^{(k)}(e) - E_j^{(k)}(e)\right|\right) \tag{16}$$

- **Chebyshev Distance**: Focuses on the maximum deviation along any embedding dimension. This conservative metric is suitable for sensitive applications, allowing aggregation only when the largest deviation between clients remains within a predefined threshold.

$$A_{ij} = \exp\left(-\max_{k=1,\ldots,d} \left|E_i^{(k)}(e) - E_j^{(k)}(e)\right|\right) \tag{17}$$

- **Jaccard Similarity**: Measures the overlap of shared entities between clients based on set intersection over union. This metric is particularly useful when embedding representations are discrete or binary, reflecting knowledge sharing at the structural level.

$$A_{ij} = \exp\left(\frac{|\mathbf{E}_i \cap \mathbf{E}_j|}{|\mathbf{E}_i \cup \mathbf{E}_j|}\right) \tag{18}$$

Once the similarity matrix $\mathbf{A}$ is constructed, personalized aggregation can be performed for each client. For a given client $i$, the personalized embedding for entity $e$ is computed as:

$$\mathbf{E}_{\text{personalized}}^{(i)}(e) = \frac{1}{Z_i(e)} \sum_{j \in \mathcal{C}_e} A_{ij} \cdot \mathbf{E}_j(e) \tag{19}$$

where the normalization factor $Z_i(e)$ is given by: $Z_i(e) = \sum_{j \in \mathcal{C}_e} A_{ij}$. To balance personalized information with local characteristics, a mixing coefficient $\alpha \in [0,1]$ can be introduced. The final embedding is computed as:

$$\mathbf{E}_{\text{final}}^{(i)}(e) = \alpha \cdot \mathbf{E}_{\text{personalized}}^{(i)}(e) + (1 - \alpha) \cdot \mathbf{E}_i(e) \tag{20}$$

The resulting embedding $\mathbf{E}_{\text{final}}^{(i)}(e)$ can then be used in subsequent steps such as standard federated averaging (*FedAvg*), serving as improved initialization for the global model.

### A.5.4 CLIENT SIDE: LOCAL FINE-TUNING

Algorithm 4 presents the client-side local fine-tuning process in the OFKGE framework. Upon receiving the distilled global embeddings from the server, each client initializes its local model and performs further optimization using its private knowledge graph data. The loss function incorporates three components: a knowledge graph embedding (KGE) loss, an embedding regularization term to preserve local-global consistency, and an optional relation-aware regularization term. By minimizing the total fine-tuning loss over several local training iterations, the client refines its embeddings to better align with local semantics while leveraging global knowledge. The resulting fine-tuned model is then prepared for downstream applications.

---

**Algorithm 4: Client-Side Local Fine-tuning**

---

**Input:** Local training rounds $T_{\text{local}}$; local knowledge graph data $D_c$; score function $f$ of selected knowledge embedding model

**Output:** Fine-tuned local model $M_{c,1}$

1 Receive distilled global embeddings: $E_{\text{global}}^c, R_{\text{global}}^c$;
2 Initialize: $E_1^c \leftarrow E_{\text{global}}^c, \quad R_1^c \leftarrow R_{\text{global}}^c$;
3 Split local dataset $D_c$ into mini-batches: $\mathcal{B} = \{b_1, b_2, \ldots, b_n\}$;
4 **for** $t = 1$ *to* $T_{local}$ **do**
5     **foreach** *batch* $b \in \mathcal{B}$ **do**
6         Compute KGE loss: $\mathcal{L}_{\text{kge}} = \sum_{\text{pos}} \sum_{\text{neg}} [\gamma + f_{\text{pos}} - f_{\text{neg}}]$;
7         Compute embedding regulation term: $\mathcal{L}_{\text{re}} = \log\left[1 + \frac{\exp(\text{sim}(E_1^c, E_0^c))}{\exp(\text{sim}(E_1^c, E_{\text{global}}^c))}\right]$;     // Eq. 8
8         Compute relation regulation term (optional): $\mathcal{L}_{\text{re-r}} = \log\left[1 + \frac{\exp(\text{sim}(R_1^c, R_0^c))}{\exp(\text{sim}(R_1^c, R_{\text{global}}^c))}\right]$;
        // usually $\gamma = 0$ to disable this term
9         Compute total loss: $\mathcal{L}_{\text{fine-tune}} = \mathcal{L}_{\text{kge}} + \beta\mathcal{L}_{\text{re}} + \gamma\mathcal{L}_{\text{re-r}}$;     // Eq. 7
10         Update embeddings: $E_1^c \leftarrow E_1^c - \eta\nabla\mathcal{L}_{\text{fine-tune}}(b), \quad R_1^c \leftarrow R_1^c - \eta\nabla\mathcal{L}_{\text{fine-tune}}(b)$
11 Construct updated local model for downstream tasks: $M_{c,1} \leftarrow (E_1^c, R_1^c)$;

---

### A.5.5 DATA GENERATION

Inspired by Wang et al. (2018), Cai & Wang (2018), and Zhao et al. (2024), the data generation method used in OFKGE integrates rules with client knowledge. By optimizing the generator, the server can synthesize auxiliary data to facilitate knowledge distillation.

Specifically, given a head–relation pair $(h, r)$, the goal of the generator is to synthesize a tail entity $t$ under rule constraints, such that the generated triple $(h, r, t)$ maintains semantic rationality and structural consistency. The input vector of the generator is defined as:

$$\text{input} = [h; r; z] \in \mathbb{R}^{2d + d_z} \tag{21}$$

where $h$ and $r$ represent the embeddings of the head entity and relation, respectively, and $z$ is a random noise vector of dimension $d_z$ introduced to enhance generation diversity. The generator $G_\theta$ consists of two fully connected layers, which map the input into the embedding of the tail entity:

$$t_{\text{gen}} = G_\theta(\text{input}) = W_2\,\sigma(W_1 \cdot \text{input} + b_1) + b_2 \tag{22}$$

To ensure the rationality of the generated entity, rule constraints are introduced to form a candidate set $C(h, r)$, and the best-performing client model under relation $r$, denoted as $f_{\text{best}}$, is selected to guide the training. For each candidate entity $j \in C(h, r)$, the similarity between its embedding $e_j$ and the generated embedding $t_{\text{gen}}$ is calculated, followed by a softmax normalization to obtain the probability distribution:

$$p_j = \frac{\exp(\text{sim}(t_{\text{gen}}, e_j)/T)}{\sum_{k \in C(h,r)} \exp(\text{sim}(t_{\text{gen}}, e_k)/T)} \tag{23}$$

where $T$ is the temperature parameter and $\text{sim}(\cdot, \cdot)$ denotes a similarity function (e.g., cosine similarity). The final tail entity is then sampled from the distribution:

$$t \sim \text{Multinomial}(p_1, p_2, \ldots, p_{|C(h,r)|}) \tag{24}$$

Since the output of the generator is a discrete entity index, following Wang et al. (2018) and Cai & Wang (2018), the training process is modeled as a reinforcement learning problem and optimized with policy gradients. The objective incorporates both semantic rationality and structural consistency. The semantic rationality is evaluated by the client model $f_{\text{best}}$, leading to the following reward function:

$$R_{\text{sim}}(h, r, t) = \tanh\left(\frac{1}{f_{\text{best}}(h, r, t)} + \epsilon\right) \tag{25}$$

The corresponding policy gradient loss is:

$$L_{\text{policy}} = \mathbb{E}_{t \sim p(\cdot|\text{input})}\left[\log p(t|\text{input}) \cdot R_{\text{sim}}\right] \tag{26}$$

To further constrain the generated embedding $t_{\text{gen}}$ and encourage alignment with the target entity embedding $e_t$, a structural loss is introduced:

$$L_{\text{struct}} = 1 - \frac{t_{\text{gen}}^T e_t}{\|t_{\text{gen}}\| \cdot \|e_t\|} \tag{27}$$

The overall loss is defined as a weighted combination of the two objectives, with a hyperparameter $\lambda$ controlling the trade-off:

$$\min_\theta L(\theta) = (1 - \lambda)L_{\text{policy}} + \lambda L_{\text{struct}} \tag{28}$$

This method relies solely on pre-extracted rules and client models, enabling the server to directly synthesize auxiliary data that supports knowledge distillation without requiring access to raw client data. Section 5.8 provides a comparison between the generated data and the real data.

### A.5.6 EXTENDS

The OFKGE framework proposed in this work focuses on efficient knowledge sharing under the one-shot setting. We leave the integration of formal privacy-preserving mechanisms, such as secure aggregation or differential privacy, to future work for ongoing improvement, which could further enhance privacy while maintaining the communication and computational benefits of the one-shot framework.

## A.6 ADDITIONAL EXPERIMENTAL SETUP AND EVALUATION

### A.6.1 DATA STATISTICS

To comprehensively evaluate our method, we conduct experiments on both multimodal and standard knowledge graph datasets. We incorporate multimodal datasets with the intention of extending

OFKGE to support heterogeneous multimodal scenarios in future work. In such settings, clients may possess different single-modality data (e.g., text, image, or numerical), leading to heterogeneous embedding mappings. Knowledge distillation, as used in OFKGE, naturally accommodates such heterogeneity by allowing the server to integrate knowledge across varying client-specific modalities. Meanwhile, to assess OFKGE's effectiveness under conventional KG embedding settings, we also include two widely used unimodal benchmark datasets. The five datasets used in our experiments are introduced in detail as follows:

- MKG-W and MKG-Y (Xu et al., 2022) are two multimodal datasets constructed from Wikidata and YAGO (Suchanek et al., 2007), respectively. Although they contain images and texts, we only use their knowledge graph structure in this study.

- DB15K (Liu et al., 2019) is a multimodal subset of DBpedia (Lehmann et al., 2015) containing images, text, and numerical information, proposed by Liu et al. (2019). Similarly, we only utilize its knowledge graph structure in our experiments.

- FB15K-237 (Toutanova et al., 2015) and NELL-995 (Xiong et al., 2017) are widely used benchmark datasets originally proposed for evaluating knowledge graph embedding methods.

Table 5: Dataset Statistics

| Dataset | #Entity | #Relation | #Triples |
|---|---|---|---|
| MKG-W | 15000 | 169 | 42746 |
| MKG-Y | 15000 | 28 | 26638 |
| DB15K | 12842 | 279 | 99028 |
| FB15K-237 | 14541 | 237 | 310116 |
| NELL-995 | 75492 | 200 | 154213 |

### A.6.2 METRICS

In the One-shot federated setting, each client communicates with the server only once and cannot continuously receive global knowledge updates, making explicit global semantic aggregation challenging. Under such severely limited communication, the key to evaluating model effectiveness lies in the performance improvement achieved by clients under the guidance of globally distilled knowledge. To this end, we adopt Mean Reciprocal Rank (MRR) and Hit@N (including Hit@1, Hit@3, and Hit@10) as primary evaluation metrics.

**MRR**  measures the model's ability to rank the correct entity (head or tail) highly when predicting the missing part of a triple. In the One-shot setting, it reflects whether integrating global knowledge improves ranking accuracy. Let $|S|$ denote the total number of test triples, and $rank_i$ be the rank assigned to the correct entity in the $i$-th triple. A higher MRR indicates a stronger tendency to rank the correct entity closer to the top. The metric is defined as:

$$MRR = \frac{1}{|S|} \sum_{i=1}^{|S|} \frac{1}{rank_i} \tag{29}$$

**Hit@N**  evaluates whether the correct entity appears in the top-N predicted results. Specifically, Hit@1 reflects the model performance in capturing global semantics, while Hit@3 and Hit@10 assess its ability to generate a useful candidate set. Using the indicator function $I(\cdot)$, which returns 1 if the condition holds and 0 otherwise, the metric is calculated as:

$$Hit@n = \frac{1}{|S|} \sum_{i=1}^{|S|} I(rank_i \leq n) \tag{30}$$

The trends of MRR and Hit@N directly reflect the model's ability to capture and integrate global semantics, and to enhance clients' knowledge completion and reasoning in communication-constrained federated environments.

### A.6.3 BASELINES

To comprehensively evaluate the effectiveness of the proposed framework in the one-shot FKGE setting, we compare OFKGE with several recent and representative baselines, including both FKGE methods and strategies inspired by one-shot learning. The compared models include FedE (Chen et al., 2021), FedR (Zhang et al., 2022b), FedM (Hu et al., 2022), FedLU (Zhu et al., 2023), PFedEG (Zhang et al., 2025), as well as two basic baselines: Single and Ensemble. FedProx (Li et al., 2020) and FedEC (Chen et al., 2022) are excluded from the comparison as they require at least two rounds of communication, making them unsuitable for the OFL setting. When constrained to a single communication round, both methods reduce to standard FedAvg, losing their distinctive corrective mechanisms and becoming functionally equivalent to FedE. Brief descriptions of each baseline are provided below:

**FedE**  It is the first framework to formalize Federated Knowledge Graph Completion (FKGC), enabling embedding learning in federated settings. It uses FedAvg to aggregate client-uploaded entity embeddings and a permutation matrix to map global entities to local spaces, preventing unauthorized access. FedE bridges federated learning and knowledge graph representation, enabling cross-silo knowledge sharing.

**FedR**  To enhance privacy, FedR improves upon FedE by aggregating relation embeddings instead of entity embeddings, leveraging the higher overlap and lower sensitivity of relations across graphs. This boosts semantic sharing and privacy protection. Experiments show FedR achieves better communication efficiency and privacy while maintaining performance comparable to FedE.

**FedM**  It is the first FKGE method that simultaneously leverages both entity and relation embeddings. To ensure privacy protection, it aggregates entity and relation embeddings on different servers. Compared to FedE and FedR, FedM demonstrates superior performance in improving local training efficiency and embedding quality.

**FedLU**  It builds on FedE, addressing data heterogeneity through Mutual Knowledge Distillation to reduce drift between local optimization and global convergence. The framework also introduces a knowledge unlearning mechanism. However, in this study, only the learning part is tested, and the forgetting module is not used.

**PFedEG**  It addresses semantic disparity in federated knowledge graphs by constructing personalized entity embeddings for each client. It builds a Client-Wise Relation Graph and aggregates entity embeddings based on this graph, improving local embedding quality. This method introduces an enhanced personalized aggregation mechanism based on FedE.

**Single**  In this setting, each client trains independently with local data while having access to the full set of entities and relations. It serves to measure the performance a client model can achieve under the ideal condition of global semantic awareness. This represents the lower-bound baseline for evaluating performance with global semantics incorporated.

**Ensemble**  Since current OFL methods are primarily designed for image classification and lack tailored approaches for knowledge graph embedding, we design a baseline that simulates one-shot average-based aggregation. After each client independently trains its local model, it uploads performance metrics on its test set, which are used as weights to aggregate models accordingly. This method mimics the one-shot model averaging strategy in image tasks and serves as a comparative baseline to validate the effectiveness of our knowledge distillation strategy. Its main limitation is that it only performs linear aggregation of embeddings, making it difficult to capture the semantic correlations of the same entity across different knowledge graphs.

A.6.4 IMPLEMENTATION DETAILS

Our OFKGE framework is implemented based on OpenKE (Han et al., 2018), a widely used open-source library for knowledge graph embedding. All experiments are conducted on a Linux server running Ubuntu 24.04.2, equipped with a single NVIDIA GeForce RTX 3070 Ti GPU (8GB VRAM). The knowledge distillation training is conducted for 1000 epochs, with both entity and relation embedding dimensions set to 200. We evaluate the model under different numbers of clients $[3, 5, 8, 10]$, and use the 3-client setting to compare its performance with baseline models and conduct ablation studies. During the server-side knowledge distillation phase, we apply a joint optimization of soft and hard losses, where the trade-off hyperparameter $\alpha$ is tuned from the set $\{0, 0.2, 0.4, 0.6, 0.8, 1\}$. Similarly, on the client side, a regularization term is introduced during the fine-tuning phase to improve the local model, with the corresponding hyperparameter $\beta$ also selected from $\{0, 0.2, 0.4, 0.6, 0.8, 1\}$. We optimize all models using the Adam optimizer (Kinga et al., 2015), and the learning rate is tuned from $\{1, 1e^{-1}, 1e^{-2}, 1e^{-3}, 1e^{-4}\}$. All baselines are reproduced using their official open-source implementations, adapted to our data splits. For consistency with the one-shot setting, we remove unrelated modules, such as the unlearning module in FedLU.

In this study, we adopt TransE as the representative knowledge graph embedding method to evaluate model performance. It should be noted that our proposed method is compatible with a wide range of KGE models (for validation experiments with RotatE, see Appendix A.6.5). Following standard evaluation protocols in KGE, we report Mean Reciprocal Rank (MRR) and Hits@N to assess link prediction performance on each client. Evaluation is conducted under the filtered setting (Bordes et al., 2013), where candidate triples appearing in training, validation, or test sets are removed during ranking.

A.6.5 EFFECT OF DIFFERENT KGE MODEL

To validate the generality and effectiveness of OFKGE beyond TransE, we further applied it to a more expressive knowledge graph embedding (KGE) model, RotatE, and evaluated it on the MKG-W, DB15K, and FB15K-237 datasets. The experimental results are shown in Table 6. As the results indicate, OFKGE outperforms the baseline methods across most metrics and datasets. This demonstrates that our framework is not only effective with the simple TransE model but also performs well with the more expressive RotatE model, indicating that OFKGE can adapt to different KGE models. These results confirm the robustness and flexibility of OFKGE.

Table 6: Valuation of RotatE on MKG-W, DB15K, and FB15K-237.

| Dataset | Method | Client 1 | | | | Client 2 | | | | Client 3 | | | |
|---|---|---|---|---|---|---|---|---|---|---|---|---|---|
| | | MRR | Hit@10 | Hit@3 | Hit@1 | MRR | Hit@10 | Hit@3 | Hit@1 | MRR | Hit@10 | Hit@3 | Hit@1 |
| MKG-W | Single | 26.51 | 34.77 | 28.37 | 21.97 | 25.85 | 34.08 | 27.47 | 21.45 | 25.07 | 22.18 | 26.81 | 20.73 |
| | Ensemble | 28.35 | 36.77 | 30.14 | 23.70 | 27.53 | 36.29 | 29.43 | 22.75 | 27.12 | 36.19 | 28.97 | 22.43 |
| | FedE | 28.92 | 37.94 | 31.03 | 24.01 | 28.12 | 37.77 | 30.23 | 23.10 | 27.61 | 36.01 | 29.43 | 22.83 |
| | FedR | 27.49 | 35.53 | 29.52 | 22.92 | 26.42 | 34.99 | 28.23 | 21.70 | 25.87 | 34.05 | 27.83 | 21.29 |
| | FedM | 28.92 | 37.51 | 30.99 | 24.04 | 28.29 | 37.28 | 30.48 | 23.28 | 27.44 | 36.61 | 29.41 | 22.48 |
| | FedLU | 28.57 | 37.94 | 30.43 | 23.66 | 28.02 | 37.44 | 30.39 | 22.91 | 27.33 | 36.05 | 29.05 | 22.58 |
| | pFedEG | 28.92 | 37.65 | 30.99 | 24.16 | 28.18 | 37.51 | 30.42 | 23.04 | 27.19 | 35.99 | 29.16 | 22.37 |
| | **Ours** | **29.99** | **38.77** | **32.12** | **25.05** | **29.42** | **39.03** | **31.78** | **24.25** | **28.73** | **37.21** | **30.49** | **24.08** |
| DB15K | Single | 26.37 | 45.06 | 32.52 | 15.99 | 26.76 | 45.29 | 32.58 | 16.54 | 26.70 | 45.03 | 32.58 | 16.49 |
| | Ensemble | 27.98 | 47.97 | 34.82 | 16.72 | 28.21 | 48.47 | 34.80 | 17.04 | 27.98 | 48.03 | 35.00 | 16.69 |
| | FedE | 27.67 | 48.11 | 35.15 | 15.92 | 28.09 | 48.08 | 35.07 | 16.75 | 27.84 | 47.92 | 34.56 | 16.62 |
| | FedR | 27.96 | 47.98 | 35.20 | 16.61 | 28.12 | 48.04 | 35.03 | 16.95 | 27.69 | 47.90 | 34.50 | 16.38 |
| | FedM | 27.86 | 48.32 | 34.76 | 16.46 | 28.22 | 48.45 | 35.41 | 16.69 | 27.92 | 47.99 | 34.78 | 16.57 |
| | FedLU | 28.10 | 48.36 | 35.26 | 16.60 | 28.13 | 48.30 | 35.02 | 16.84 | 27.99 | 48.07 | 34.65 | 16.81 |
| | pFedEG | 28.09 | 48.24 | 35.35 | 16.72 | 28.14 | 48.21 | 35.18 | 16.76 | 27.75 | 47.66 | 34.76 | 16.32 |
| | **Ours** | **29.52** | **49.92** | **37.22** | **17.68** | **29.75** | **50.13** | **37.15** | **18.06** | **29.45** | **49.73** | **36.64** | **17.83** |
| FB15K-237 | Single | 31.88 | 55.99 | 38.23 | **19.42** | 32.05 | 55.91 | 38.40 | **19.65** | 31.60 | 55.38 | 37.82 | **19.28** |
| | Ensemble | 30.74 | 55.10 | 36.54 | 18.50 | 31.11 | 55.28 | 37.01 | 18.83 | 30.69 | 54.82 | 36.70 | 18.42 |
| | FedE | 30.94 | 55.12 | 36.95 | 18.67 | 31.13 | 55.11 | 37.26 | 18.80 | 30.73 | 54.60 | 36.71 | 18.48 |
| | FedR | 30.85 | 54.98 | 36.89 | 18.53 | 31.25 | 55.07 | 37.47 | 18.94 | 30.76 | 54.55 | 36.78 | 18.52 |
| | FedM | 30.91 | 55.12 | 37.09 | 18.46 | 31.01 | 55.08 | 37.23 | 18.64 | 30.73 | 54.62 | 36.61 | 18.55 |
| | FedLU | 30.91 | 55.18 | 36.86 | 18.62 | 31.09 | 55.17 | 37.09 | 18.83 | 30.84 | 54.86 | 36.77 | 18.62 |
| | pFedEG | 30.77 | 55.01 | 36.86 | 18.38 | 31.07 | 55.10 | 37.05 | 18.78 | 30.83 | 54.72 | 36.86 | 18.54 |
| | **Ours** | **31.88** | **56.63** | **38.45** | 19.13 | **32.24** | **56.74** | **38.74** | 19.59 | **31.81** | **56.24** | **38.03** | 19.20 |

### A.6.6 EFFECT OF REGULARIZATION TERM

To validate the effectiveness of the regularization term (Eq. 8) during the client side fine-tuning phase, we compared it with several alternative designs, including L2 regularization, KL divergence, and cosine similarity, on the MKG-W, DB15K, and FB15K-237 datasets. In our experiments, cosine similarity is defined as $1 - cos(E_{global}, E_{local})$, measuring the dissimilarity between local and global entity embeddings. The detailed results are reported in Table 7. As shown, our proposed regularization consistently outperforms the alternatives across almost all metrics. Although L2 regularization and KL divergence provide some regularization effect, their performance is generally lower than our design, likely due to their limited ability to capture semantic consistency between local and global embeddings. Cosine similarity occasionally outperforms L2 regularization and KL divergence but exhibits instability across different datasets and clients. This is likely because cosine similarity emphasizes directional alignment rather than comprehensive semantic consistency, leading to fluctuating performance in heterogeneous settings. Overall, our design demonstrates clear advantages in both performance and stability.

Table 7: Comparison of Different Regularization Term on MKG-W, DB15K, and FB15K-237.

| Dataset | Method | Client 1 | | | | Client 2 | | | | Client 3 | | | |
| --- | --- | --- | --- | --- | --- | --- | --- | --- | --- | --- | --- | --- | --- |
| | | MRR | Hit@10 | Hit@3 | Hit@1 | MRR | Hit@10 | Hit@3 | Hit@1 | MRR | Hit@10 | Hit@3 | Hit@1 |
| MKG-W | L2 | 20.53 | 38.83 | 27.19 | 9.98 | 20.51 | 38.78 | 27.28 | 9.99 | 19.02 | 37.42 | 25.43 | 8.59 |
| | KL | 20.61 | 38.11 | 27.50 | 10.07 | 20.32 | 37.79 | 27.24 | 9.81 | 18.79 | 36.92 | 25.72 | 8.22 |
| | Cosine | 20.67 | 39.10 | 27.69 | 9.92 | 20.38 | 38.82 | 27.28 | 9.66 | 19.04 | 37.15 | 25.62 | 8.76 |
| | **Ours** | **21.81** | **40.61** | **29.34** | **10.52** | **21.50** | **40.62** | **29.18** | **9.97** | **20.59** | **39.81** | **28.29** | **9.02** |
| DB15K | L2 | 24.10 | 47.75 | 33.04 | 10.44 | 24.54 | 48.08 | 33.66 | 10.81 | 24.15 | 47.57 | 33.62 | 10.28 |
| | KL | 23.82 | 47.50 | 32.81 | 10.06 | 24.32 | 48.01 | 33.68 | 10.45 | 24.26 | 47.59 | 33.86 | 10.34 |
| | Cosine | 23.84 | 47.65 | 32.90 | 9.98 | 24.46 | 47.98 | 33.66 | 10.80 | 24.10 | 47.41 | 33.64 | 10.26 |
| | **Ours** | **24.95** | **49.68** | **34.52** | **10.59** | **25.50** | **49.99** | **34.93** | **11.28** | **25.10** | **49.07** | **34.57** | **10.92** |
| FB15K -237 | L2 | 30.77 | 56.64 | 37.72 | 17.42 | 31.23 | 57.01 | 38.19 | 17.91 | 30.84 | 56.11 | 37.79 | 17.61 |
| | KL | 30.94 | 56.62 | 37.95 | **17.66** | 31.18 | 56.91 | 38.37 | 17.80 | 30.76 | 56.14 | 37.82 | 17.47 |
| | Cosine | 30.80 | 56.62 | 37.73 | 17.48 | 31.17 | 57.02 | 38.22 | 17.80 | 30.86 | 56.19 | 37.85 | 17.61 |
| | **Ours** | **31.03** | **57.05** | **38.17** | 17.55 | **31.39** | **57.18** | **38.66** | **17.95** | **31.09** | **56.58** | **38.14** | **17.75** |

### A.6.7 EVALUATION ON KNOWLEDGE GRAPHS WITH LOW OVERLAP

To evaluate the effectiveness of OFKGE when client knowledge graphs originate from different domains with minimal overlap, we selected two real-world datasets: MKG-W and FB15K-237. By performing entity matching to identify the same entities across these datasets, we calculated an entity overlap of only 8.15% (see Table 5 for detailed statistics of datasets). The experiment involved two clients: Client 1 owns MKG-W, and Client 2 owns FB15K-237. The experimental results are presented in Table 8. From the results, it can be observed that even when the client knowledge graphs are from largely different domains and have low overlap, OFKGE still outperforms other methods across most metrics. This indicates that the ASA mechanism can effectively capture semantic information across clients from different domains, enabling knowledge sharing and aggregation of global embeddings. The experiment validates the robustness and effectiveness of OFKGE in low-overlap, multi-domain scenarios and demonstrates the framework's strong adaptability.

Table 8: Performance Comparison of OFKGE on Low-overlap Client Knowledge Graphs.

| Method | Client 1 (MKG-W) | | | | Client 2 (FB15K-237) | | | |
| --- | --- | --- | --- | --- | --- | --- | --- | --- |
| | MRR | Hit@10 | Hit@3 | Hit@1 | MRR | Hit@10 | Hit@3 | Hit@1 |
| Single | 21.02 | 39.97 | 29.30 | 9.23 | 29.50 | 48.49 | 33.37 | 19.79 |
| Ensemble | 23.23 | 41.76 | 30.32 | 12.22 | 30.01 | 49.14 | 33.80 | 20.35 |
| FedE | 23.35 | 42.36 | 30.34 | 12.24 | 29.95 | 49.05 | **33.88** | 20.23 |
| FedR | 21.69 | 40.31 | 28.97 | 10.60 | 29.69 | 48.51 | 33.51 | 20.14 |
| FedM | 23.66 | 42.25 | 30.28 | **12.93** | 29.96 | 49.19 | 33.76 | 20.24 |
| FedLU | 23.58 | 42.20 | 30.20 | 12.69 | 30.03 | 49.28 | 33.87 | 20.33 |
| **OFKGE** | **23.75** | **42.74** | **30.86** | 12.48 | **30.19** | **49.34** | 33.87 | **20.50** |

### A.6.8 EFFECTIVENESS OF SYNTHETIC DATA

During the knowledge distillation process, we leverage synthetic data to facilitate learning. The training uses a combination of soft and hard labels rather than relying solely on the absolute correctness of each triple. This allows the distillation to capture overall relational patterns rather than overfitting individual triples. Consequently, moderate semantic deviations or occasional factual errors in the synthetic data have limited impact on the global knowledge. Existing studies on knowledge distillation using synthetic data, such as Zhang et al. (2022a); Dai et al. (2024); Zeng et al. (2024), also support its effectiveness. Moreover, Experimental results (Table 1) demonstrate that using synthetic data for server-side knowledge distillation consistently improves the performance of local models, outperforming other baseline methods, highlighting the practical value of synthetic data in OFKGE.

Furthermore, We observed that key hyperparameters in the synthetic data generation process, such as $\lambda$ (see Equation 28 in Appendix A.5.5), can affect the quality of the synthetic data, which in turn indirectly influences the distillation performance of OFKGE. The overall performance under different $\lambda$ settings on the FB15K-237 dataset are shown in Figure 5, where the reported metrics are averages across all clients. The results indicate that OFKGE achieves the best performance at $\lambda = 0.2$, suggesting that a moderate setting of this hyperparameter provides an optimal balance in the synthetic data generation process. Values that are too small ($\lambda = 0.0$) or too large ($\lambda \geq 0.4$) slightly reduce performance, indicating that both insufficient and excessive weighting can negatively affect the quality of the synthetic data.

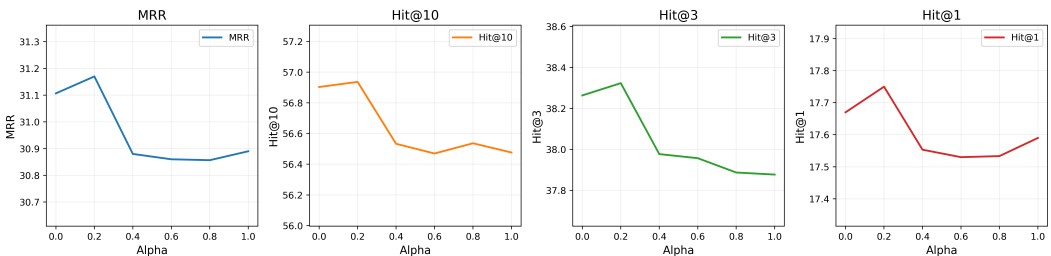

Figure 5: Comparison of different $\lambda$ settings on FB15K-237

### A.6.9 EFFECT OF NUMBER OF CLIENTS

To further evaluate the scalability of OFKGE in more realistic federated settings, we conducted experiments with 30, 50, 80, and 100 clients on FB15K-237. The results are summarized in Figure 6. Across all settings, OFKGE consistently outperforms the Single in MRR, Hit@3, and Hit@1, and maintains competitive Hit@10 performance as the number of clients increases. The observed increase in MRR, Hit@3, and Hit@1 can be attributed to the richer candidate set of clients, allowing the ASA mechanism to select more suitable teachers for each batch and provide more accurate guidance. The slight decrease in Hit@10 is likely due to the finer data partitioning, which disperses some rare entities across clients and slightly reduces coverage at lower ranks. These findings demonstrate that the proposed ASA mechanism remains effective even as the number of clients scales up, confirming the robustness of our approach under large-scale scenarios.

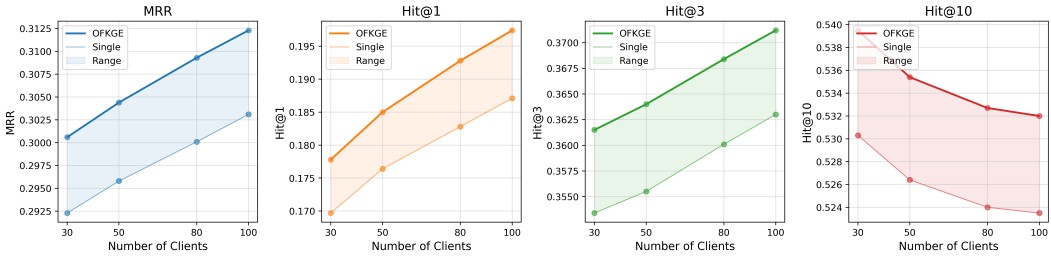

Figure 6: Performance under different numbers of clients on FB15K-237

### A.6.10 CASE STUDY

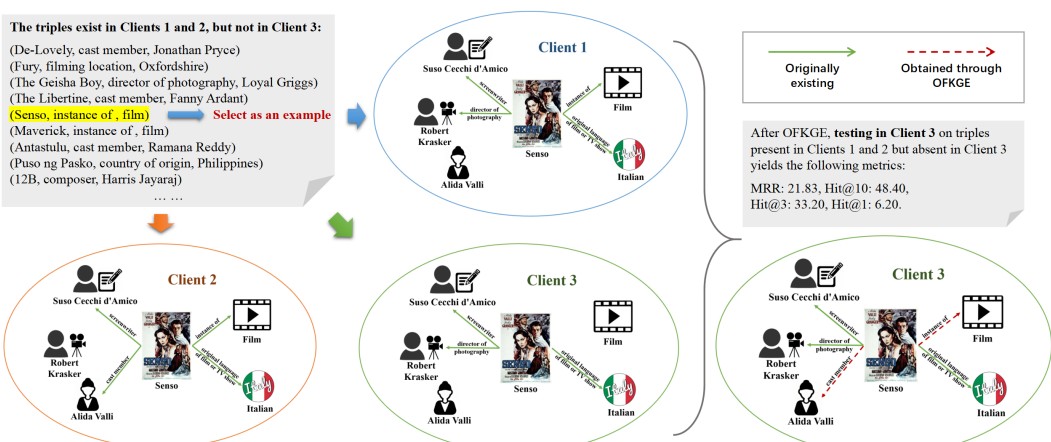

Figure 7: Case Study of a Randomly Selected Triple found in Clients 1 and 2 but missing from 3

To further demonstrate the effectiveness of OFKGE in enabling knowledge sharing across clients in a federated knowledge graph scenario, we present a representative case centered on the film Senso, as shown in Figure 7. This example illustrates how OFKGE allows clients with incomplete knowledge to benefit from the rich semantics held by others, improving local inference capabilities.

We consider three clients: Client 1, Client 2, and Client 3. Several knowledge triples, such as (De-Lovely, cast member, Jonathan Pryce) and (Fury, filming location, Oxfordshire), are available in Clients 1 and 2 but are absent in Client 3. Among these, the triple (Senso, instance of, film) is selected for detailed analysis. Before applying OFKGE, Clients 1 and 2 already possess comprehensive knowledge about Senso, including relations like screenwriter (Suso Cecchi d'Amico), director of photography (Robert Krasker), cast member (Alida Valli), instance of (film), country of origin (Italy), and original language (Italian). These facts form a well-connected semantic graph within their local models. In contrast, Client 3 initially lacks critical knowledge, including the (instance of, film) relation, which limits its reasoning capacity regarding the entity Senso.

After training with OFKGE, Client 3 successfully acquires the missing (instance of, film) relation, demonstrating that the framework can effectively transfer relational knowledge from clients with complete data to those with limited information. This transfer occurs without direct access to private data, highlighting the privacy-preserving nature of OFKGE.

To quantitatively evaluate this enhancement, we assess Client 3's performance on a set of triples that were originally absent from its local graph but present in the other clients. The evaluation results show that the local model achieves the MRR of 21.83, Hit@10 of 48.40, Hit@3 of 33.20, and Hit@1 of 6.20. These results indicate that OFKGE not only enables access to missing knowledge but also significantly enhances the downstream inference performance of the client.

This case study clearly demonstrates the ability of OFKGE to bridge semantic gaps among clients in federated environments. By transmitting relation-specific distilled knowledge, clients can effectively integrate global semantic information into their local models. Such capability is particularly beneficial for downstream applications such as large-scale question answering systems and Internet of Things (IoT) scenarios, where local devices require access to generalized global knowledge to support accurate and context-aware reasoning.

### A.7 LIMITATIONS

Although OFKGE strikes a good balance between communication efficiency, computational cost, and reasoning performance, it still has certain limitations, primarily in the following two aspects:

**Limited adaptability to dynamic environments**   OFKGE adopts a one-shot communication mechanism, which significantly reduces communication overhead but also limits flexibility. In practical scenarios, knowledge graphs are often dynamic, with new entities and relations continuously emerging. However, OFKGE relies on a static modeling approach that cannot effectively capture these changes in a timely manner. As a result, its applicability is limited in long-term deployments, real-time systems, or open environments where knowledge evolves rapidly.

**Dependence on the quality of client models for distillation**   The effectiveness of server-side knowledge distillation in OFKGE is highly dependent on the quality of the client models that serve as teachers. If some clients have low-quality, biased, or overfitted models due to noisy data, insufficient training, or limited resources, this can negatively impact the global distillation process and reduce the robustness and generalization ability of the final model. Furthermore, OFKGE assumes that each client has access to sufficient and high-quality training data. In practice, however, clients often face challenges such as data sparsity, class imbalance, or missing labels. These issues are particularly problematic in scenarios with highly skewed or long-tailed data distributions and may significantly degrade overall system performance.

