# OpenReview forum: "Reimagining Federated Knowledge Graph Embedding with One-Shot Adaptive Semantic Alignment"
_ICLR.cc/2026/Conference — Submitted to ICLR 2026_

### Official Review · Reviewer_jtAW · 2025-10-19

**Soundness:** 2
**Presentation:** 3
**Contribution:** 1
**Rating:** 2
**Confidence:** 4

**Summary:**

The paper utilizes one-shot federated learning paradigms (OFL) to scenarios in collaborated training of knowledge graph embeddings. The training procedure operates as a three-stage process: Starting with a somewhat conventional client side upload, the server then use a dynamic teacher selection with picks the most powerful client-side model that guides the server side training of a global model. The procedure is finished by a client-side fine-tuning leveraging the server-side global model. Experimental investigations show promising performance of the proposed OFKGE method

**Strengths:**

- The authors identified the communication overhead and privacy leakage as an important challenge in contemporary developments in federated knowledge graph embedding. One-shot FL might be a potential solution to this challenge.
- The overall writing of the paper is clear

**Weaknesses:**

- Limited novelty: While the authors claimed OFKGE to be the first attempt in applying OFL methodology to federated KGE problems, the adaptation of OFL seems pretty trivial for me. The primary contribution of OFKGE seems to be the multi-teacher distillation method used in the second stage of the training pipeline (server-side distillation). However, such kinds of procedures have been previously studied in federated learning context such as [1], while [1] did not specifically handles OFL problems, I believe such kinds of extensions is pretty straightforward. Moreover, the so-called **adaptive teacher selection** method as detailed in equation (6) in the paper seems to be a *dynamic* selection method instead of an *adaptive* selection method, as it merely uses client-side uploaded weights to determine the per-step best teacher, but no statistics of previous server-side training are deviced------which for me would then indicate a formal sense of adaptivity.
- Privacy guarantees: As the authors also pointed out in the intro section that privacy leakage of FKGE is a significant concern [2], it would be then interesting to investigate the actual privacy protection level of OFKGE. While it seems that OFKGE does not utilize strong privacy protocols like DP, it should provide empirical evidence of protection. I think in addition to performance reports, the authors should also state the privacy protection capabilities of OFKGE.

**Questions:**

In addition to the weaknesses, I have one additional question:
- **On relation-aware performance matrices**: During the first stage of OFKGE, the clients commute *relation-aware performance matrices* using their local datasets and the server uses this metric to guide the distillation process in the second stage of OFKGE. Therefore it makes me wonder that whether the distributional shifts of client-local datasets may severely affect the performance metrics------As the server do NOT have access to those evaluation datasets, it seems that relying on such types of matrices is somewhat unreliable. Please elaborate more on this design choice.


[1]. Li, Mingyi, et al. "Resource-aware federated self-supervised learning with global class representations." Advances in Neural Information Processing Systems 37 (2024): 10008-10035.
[2]. Hu, Yuke, et al. "Quantifying and defending against privacy threats on federated knowledge graph embedding." Proceedings of the ACM Web Conference 2023. 2023.

---

> ### Author Response · Authors · 2025-11-22
> **Response to Reviewer jtAW (Part 1)**
>
> Thank you for the valuable comments. Below, we have carefully addressed each of your comments.
>
> > **`W1`**: We agree that multi-teacher distillation is a key component of OFKGE, but we argue that its value lies in **designing a one-shot federated learning framework tailored for KGE**.
> >
> > **`W1.1`**: The one-shot paradigm itself is flexible and can be adapted to different domains, yet **each domain poses unique challenges**. Just as federated learning (FL) encounters domain-specific difficulties, one-shot FL methods also need to address issues specific to the data modality. For instance, [1] studies FL on image data, while [2] explores it on KGs, each proposing solutions suitable for their respective domains. Traditional one-shot federated learning methods mainly target image data [3, 4], including the work you mentioned [5]. However, **KGs present distinct characteristics and challenges** compared to image data, requiring specific method design. Notably, one-shot FL approaches for images often do not require alignment or fine-tuning [6, 7], which contrasts with the challenges we face in KGs. Our multi-teacher distillation leverages ASA for semantic alignment and incorporates client-side fine-tuning to balance global knowledge guidance with local data adaptation, enabling enhanced KGE under a single communication round. Therefore, **directly transferring existing image-based methods to KGE is not applicable**.
> >
> >**`W1.2`**: In this work, the so-called **adaptive teacher selection** refers to the **dynamic selection of the optimal teacher for each batch**. Its *adaptiveness* lies in the fact that the teacher choice adjusts according to the entities and relations present in the batch, rather than relying on fixed rules. Although the current method does not use server-side historical statistics, the relation-level performance matrices uploaded by clients already reflect each client’s relative competence across different relations. Therefore, for each teacher selection, the client that performs best on the current batch is adaptively chosen, achieving **batch-level, relation-aware dynamic adaptation**. In other words, the *adaptiveness* of our method primarily manifests as locally optimal selection, ensuring an efficient and stable distillation process while reducing computational and communication overhead.
>
> > **`W2`**: We would like to clarify and emphasize the following points:
> >
> > - **One-shot federated learning offers inherent privacy advantages.** Our framework builds upon the inherent privacy-preserving nature of federated learning, which avoids direct data sharing by design. Furthermore, the adoption of a one-shot communication scheme provides **additional privacy benefits, particularly in mitigating attacks such as gradient inversion attacks**. Since our method avoids iterative gradient exchanges altogether, the risk of reconstructing local training data from shared updates, as highlighted in works such as [8, 9, 10], is significantly reduced. The fact that one-shot federated learning avoids the transmission of intermediate gradients or model updates during training has been demonstrated in prior literature [7, 11, 12] to offer stronger resistance to inversion and reconstruction attacks compared to multi-round schemes.
> > - **Our focus is on framework design rather than privacy guarantees**. The main contribution of this work lies in designing an efficient FKGE framework that enables knowledge sharing under a one-shot setting. Importantly, the one-shot design of OFKGE significantly **reduces communication overhead**, as it requires only a single round of interaction between clients and the server. While we acknowledge the importance of formally quantifying privacy (e.g., via differential privacy), this is beyond the scope of the current work. As stated in ***Appendix A.5.7***, *“The OFKGE framework proposed in this work focuses on efficient knowledge sharing under the one-shot setting. We leave the integration of formal privacy-preserving mechanisms, such as secure aggregation or differential privacy, to future work for ongoing improvement.”*

---

> ### Author Response · Authors · 2025-11-22
> **Response to Reviewer jtAW (Part 2)**
>
> > **`Q1`: On relation-aware performance matrices**
>
> > - **`Q1.1`**:  Although client data may exhibit distributional shifts, such differences **do not severely affect the reliability of the relation-aware performance matrices**. On the contrary, the relation-aware performance matrices and the teacher selection mechanism in OFKGE are inherently intended to remain robust under such heterogeneous settings. For example, a common scenario in FKGE is that different clients hold different relation types or highly imbalanced relation data. A client may contain abundant triples for certain relations but very few for others. Such a shift does not weaken our performance matrices; instead, it is **directly reflected** in them. Since each client model naturally performs better on relations with sufficient data and worse on relations with scarce data, this performance discrepancy is accurately captured by the relation-aware performance matrix. During the second-stage distillation, we first identify candidate clients based on the entities appearing in the current batch, and then select from these candidates the best-performing client on the involved relations according to its relation-aware performance matrices. This ensures that each client is **only chosen as the teacher for the relations in which it excels**, preventing it from negatively influencing the global model on relations where its performance is weak. Thus, the relation-aware performance matrix not only reliably supports effective teacher selection, but these distributional shifts are also effectively managed and leveraged by our design to achieve robust and efficient distillation.
> >
> > - **`Q1.2`**:  This design choice is motivated by the need for relation-aware performance matrices to **help the server perform teacher selection**, and uploading relation matrices instead of evaluation data **preserves privacy**, since evaluation data may contain sensitive information. By using relation matrices, we can select the best teacher for each batch in a principled manner. In knowledge graphs, **relations connect entities and are often more important than individual entities**[13, 14], and KGE fundamentally aims to learn relational patterns. Therefore, using relations as the basis for teacher selection allows the server to identify, for each batch, the client that is relatively most proficient in the corresponding relation, **transferring the relation-level knowledge from the client’s local model to the global model**. Intuitively, this is akin to a student finding the most suitable teacher for guidance (locally optimal) in each round, resulting in more effective learning. The effectiveness of this design is further validated by our experiments. As shown in ***Table 1***, compared to baseline models, our method achieves significant performance improvements, demonstrating the effectiveness of relation-level teacher selection.
>
> We hope the above rebuttal resolves your concerns. If you still have questions, we are willing to answer your further questions.
>
> ***Reference***:
>
> > [1] Li, Q., He, B., & Song, D. (2021). Model-contrastive federated learning. *CVPR*.
> >
> > [2] Zhu, X., Li, G., & Hu, W. (2023). Heterogeneous federated knowledge graph embedding learning and unlearning. *WWW*.
> >
> > [3] Zhang, J., Chen, C., Li, B., et al.. (2022). Dense: Data-free one-shot federated learning. *NeurIPS*.
> >
> > [4] Diao, Y., Li, Q., & He, B. (2023). Towards addressing label skews in one-shot federated learning. *ICLR*.
> >
> > [5] Li, M., Zhang, X., Wang, Q., et al.. (2024). Resource-aware federated self-supervised learning with global class representations. *NeurIPS*.
> >
> > [6] Zeng, H., Xu, M., Zhou, T., et al.. (2024). One-shot-but-not-degraded Federated Learning. *MM*.
> >
> > [7] Bai, J., Song, Y., Wu, D., et al.. (2025). A unified solution to diverse heterogeneities in one-shot federated learning. *KDD*.
> >
> > [8] Hatamizadeh, A., Yin, H., Molchanov, P., et al. (2023). Do gradient inversion attacks make federated learning unsafe?. *IEEE Transactions on Medical Imaging*.
> >
> > [9] Huang, Y., Gupta, S., Song, Z., et al. (2021). Evaluating gradient inversion attacks and defenses in federated learning. *NeurIPS*.
> >
> > [10] Geng, J., Mou, Y., Li, Q., et al. (2023). Improved gradient inversion attacks and defenses in federated learning. *IEEE Transactions on Big Data*.
> >
> > [11] Mendieta, M., Sun, G., & Chen, C. (2025). Navigating heterogeneity and privacy in one-shot federated learning with diffusion models. *WACV*.
> >
> > [12] Liu, X., Liu, L., Ye, F., Shen, et al. (2024). Fedlpa: One-shot federated learning with layer-wise posterior aggregation. *NeurIPS*.
> >
> > [13] Ji, S., Pan, S., Cambria, E., et al. (2021). A survey on knowledge graphs: Representation, acquisition, and applications. *IEEE transactions on neural networks and learning systems*.
> >
> > [14] Yue, L., Zhang, Y., Yao, Q., et al. (2023). Relation-aware Ensemble Learning for Knowledge Graph Embedding. *EMNLP*.

---

> > ### Comment · Reviewer_jtAW · 2025-11-22
> > **Reviewer response**
> >
> > First I would like to thank the authors for their rebuttal effort.
> >
> > After reading the rebuttal, I still do not think my primary concerns are addressed: The core contribution of this paper is to generalize an idea that has been developed in the federated learning community to FKGE scenario. This could be some scientific advancement itself, but it indeed appears to me that this seems to be a combination of contemporary developments. As also pointed out by reviewer XiuL, the paper also lacks discussion on theoretical side of the problem (either from an optimization viewpoint or from formal privacy viewpoint) which might instead be an important contribution.
> > I appreciate your effort, but I will keep my score.

---

> > > ### Author Response · Authors · 2025-11-25
> > >
> > > Thank you for your careful reading and comments.
> > >
> > > As recognized by reviewers `jYhd`, `jcKT`, and `XiuL`, **our work makes a meaningful contribution**, and we are the **first to bring OFL into the KGE scenario**.  Federated learning is fundamentally a training paradigm applied across various domains. If interpreted strictly as *simple transfer*, then essentially all FL papers could be considered as such. This perspective, however, overlooks the **unique challenges arising from the specific data characteristics and structures in knowledge graphs**.
> > >
> > > While we appreciate your evaluation, we respectfully maintain that **it does not entirely reflect the novelty and the unique challenges that our work addresses**. In particular, regarding the concerns raised by reviewer `XiuL`, we have provided detailed explanations to justify the rationale and effectiveness of our approach, as summarized below:
> > >
> > > > **`W2`**: The paper lacks theoretical justification for why one-shot learning should work well for KGE. There is no convergence analysis or theoretical bounds on the performance gap between one-shot and multi-round approaches.
> > > >
> > >
> > > > **`Response`**: We have included relevant analysis in ***Appendix A.4.1 (line 873 - 898)***. The details are provided below.
> > > >
> > > >**`W2.1`**: The core idea of the one-shot federated learning approach we use is to **extract structured knowledge from each client’s local KGE model via knowledge distillation and transfer it to the global student model on the server in a single-shot communication**. The knowledge contained in a knowledge graph can be transmitted through the distillation process [1,2,3]. Specifically, assuming that the local KGs of all clients collectively form a global KG, each client can only observe a subset of the global KG, i.e., a local subgraph. Consequently, the client model captures the semantic and reasoning capabilities of this local subgraph, and the knowledge it encodes can be understood as the **semantic patterns between entities and relations** within the subgraph, which we refer to as ***knowledge***. On the server side, **knowledge distillation allows these local *knowledge* fragments from all subgraphs to be integrated into a global student model**. Based on this mechanism, extracting KG knowledge from client models via knowledge distillation and integrating it to form a global model is both reasonable and effective.
> > > >
> > > > **`W2.2`**: In our **one-shot** approach, the global model converges through **knowledge distillation on the server side**. Specifically, we first initialize the global model using an aggregation method such as FedAvg (other initialization strategies are detailed in ***Appendix A.5.3***), and then gradually acquire knowledge from client local models by optimizing the knowledge distillation objective.
> > > > $$
> > > > \mathcal{L} _ {distillation}= \alpha \mathcal{L} _ {kge} + (1-\alpha)\mathcal{L} _ {distill-soft}
> > > > $$
> > > > In each training step, operations such as gradient descent are applied to **minimize this objective on server**, allowing the student model to progressively approach the selected teacher models.
> > > >
> > > > In contrast, **multi-round** approaches rely on **iterative client-server interactions** to gradually achieve global convergence. Existing FKGE methods [3,4] mainly perform FedAvg on the server side (e.g., ***Eqn 13***) and **minimize the KGE objective on clients** in each communication round.
> > > > $$
> > > > \mathcal{L} _ {kge} = \sum _ {(h,r,t) _ {\text{pos}}} \sum _ {(h', r, t') _ {\text{neg}}} \left[\gamma + f(h, r, t) - f(h', r, t')\right] _ +
> > > > $$
> > > > If multi-round methods are naively adapted to a one-shot setting, the server would only perform a single FedAvg, which is insufficient to fully transfer client knowledge.
> > > >
> > > > Therefore, our one-shot method addresses the issue that simple aggregation in multi-round methods cannot fully capture the diverse knowledge from clients in a one-shot scenario. After aggregation, knowledge distillation is applied to effectively transfer client knowledge to the global model, enabling it to **integrate more knowledge** than traditional aggregation methods.
> > >
> > > We sincerely thank you once again for your careful reading and feedback.
> > >
> > > ***Reference***
> > > >[1] Liu, J., Wang, P., Shang, Z., et al. (2023). Iterde: an iterative knowledge distillation framework for knowledge graph embeddings. *AAAI*.
> > > >
> > > >[2] Liu, Y., Sun, Z., Li, G., & Hu, W. (2022). I know what you do not know: Knowledge graph embedding via co-distillation learning. *CIKM*.
> > > >
> > > >[3] Zhu, X., Li, G., & Hu, W. (2023). Heterogeneous federated knowledge graph embedding learning and unlearning. *WWW*.
> > > >
> > > >[4] Chen, M., Zhang, W., Yuan, Z., et al. (2021). Fede: Embedding knowledge graphs in federated setting. *IJCKG*.

---

> > > ### Author Response · Authors · 2025-11-26
> > >
> > > As a further clarification and extension of our previous response, we would like to emphasize that **our approach is not a simple *direct migration* of OFL to the KGE scenario**, and its contribution lies specifically in designing a one-shot federated learning framework tailored to the unique challenges of knowledge graph embedding.
> > >
> > > - **FL and OFL in different scenarios**: **`FL`** is indeed a general training paradigm applied across domains. For example, [1,2] for images, [3] for graph data, [4] for knowledge graphs, and [5] for multimodal data. **These works are not trivial and introduce solutions tailored to the specific challenges of their respective data environments**. Similarly, **`OFL`**, as a recently proposed training paradigm [6], is characterized by constructing a global model with only a single communication round. Existing research has developed various methods tailored to different scenarios, such as [7] for images, [8] which first introduces OFL into graph learning (GL), and [9] for multimodal data. However, **to date, no study has applied OFL to KGE**.
> > >
> > > - **Unique challenges of KG**: As highlighted in prior work [10], which addresses unsolved challenges specific to GL: "*Current OFL approaches are primarily designed for traditional image data and fail to capture the fine-grained structural information of local graph data.*" Similarly, **applying OFL to KGE also faces entirely new challenges**, as the data characteristics of knowledge graphs differ substantially from other modalities such as images and homogeneous graphs. For **`image data`**, OFL relies on CNN- or Transformer-extracted features [1,2], where server-side aggregation only averages model weights because client feature spaces are naturally aligned, **requiring no semantic alignment**. In contrast, KGE captures structured semantic patterns between entities and relations. Simple weight aggregation fails to capture these rich semantic patterns. For **`GL`**, typical graphs are homogeneous with uniform node and edge types, focusing on node features and local neighborhoods for tasks such as node classification or node representation learning[3,8,10]. Since the semantic patterns are relatively uniform, there is **no need to handle diverse semantic patterns**. Knowledge graphs, however, are heterogeneous, with multiple entity and relation types in triples encoding distinct semantics. Directly applying GL-based OFL cannot transfer these diverse semantic patterns effectively.
> > >
> > > To address these challenges, we propose the **ASA mechanism**, which performs relation-aware, fine-grained adaptive semantic alignment, effectively transferring local semantic patterns to the global model. Thus, **existing OFL methods for images, graphs, or multimodal tasks cannot be directly applied, and our work provides the first solution that enables OFL to function effectively for KGE**.
> > >
> > > We hope that our explanation clarifies the novelty and significance of our approach.
> > >
> > > ***Reference***:
> > > > [1] Wang, Z., Wang, Z., Fan, X., et al. (2025). Federated Learning with Domain Shift Eraser. *CVPR*.
> > > >
> > > >[2] Yang, X., Huang, W., & Ye, M. (2024). Fedas: Bridging inconsistency in personalized federated learning. *CVPR*.
> > > >
> > > >[3] Wan, G., Huang, W., & Ye, M. (2024). Federated graph learning under domain shift with generalizable prototypes. *AAAI*.
> > > >
> > > >[4] Zhu, X., Li, G., & Hu, W. (2023). Heterogeneous federated knowledge graph embedding learning and unlearning. *WWW*.
> > > >
> > > >[5] Chen, H., Zhang, Y., Krompass, D., et al. (2024). Feddat: An approach for foundation model finetuning in multi-modal heterogeneous federated learning. *AAAI*.
> > > >
> > > >[6] Guha, N., Talwalkar, A., & Smith, V. (2019). One-shot federated learning. *arXiv preprint arXiv:1902.11175*.
> > > >
> > > >[7] Bai, J., Song, Y., Wu, D., et al. (2025). A unified solution to diverse heterogeneities in one-shot federated learning. *KDD*.
> > > >
> > > >[8] Qian, J., Wan, G., Huang, W., et al. (2025). GHOST: Generalizable One-Shot Federated Graph Learning with Proxy-Based Topology Knowledge Retention. *ICML*.
> > > >
> > > >[9] Wang, N., Deng, Y., Fan, S., et al. (2025). Multi-Modal One-Shot Federated Ensemble Learning for Medical Data with Vision Large Language Model. *arXiv preprint arXiv:2501.03292*.
> > > >
> > > >[10] Wan, G., Qian, J., Huang, W., et al. (2025). OASIS: One-Shot Federated Graph Learning via Wasserstein Assisted Knowledge Integration. *NeurIPS*.

---

### Official Review · Reviewer_XiuL · 2025-10-30

**Soundness:** 3
**Presentation:** 3
**Contribution:** 2
**Rating:** 4
**Confidence:** 3

**Summary:**

This paper introduce a novel one-shot federated leanring framework for knowledge graph embedding to balance performance gain with communication/computation cost. The proposed methd, named OFKGE, is evaluated on 5 datasets to show the effectivenss.

**Strengths:**

1. Generally, this paper is easy to follow.

2. The authors focuses on an important problem in federated learning, reducing comunication cost while mainting performance.

**Weaknesses:**

1. The authors states that this is the first OFL framework tailored for knowledge graph embedding. However, to my knoweledge, some studies in this field have emerged. It would be better for the author to distinguish and compare the proposed method with existing ones. For example:

[1] Personalized One-shot Federated Graph Learning for Heterogeneous Clients.

In addition, existing years have witnessed a great development of one-shot learning. Some of them are desgined for federated learning setting. It is beneficial to adapt them methods in KGE to fully validate the effectivenss of the proposed method.

2. The paper lacks theoretical justification for why one-shot learning should work well for KGE. There is no convergence analysis or theoretical bounds on the performance gap between one-shot and multi-round approaches.

3. All experiments use only 3 clients by default, which is quite limited for federated settings. Figure 3 shows results up to 7 clients, but real federated scenarios may involve hundreds or thousands of clients.

4. The method requires either unlabeled data or synthetic data generation capability on the server, which may not always be available or practical.

5. The teacher selection (Eq. 4) is greedy and based on relation-wise performance. What if different clients excel at different entity types rather than relations?

**Questions:**

1. How sensitive is OFKGE to the quality of synthetic data?

2. How were multi-round baselines (FedE, FedLU, etc.) adapted to the one-shot setting? Did you simply run them for one round, or were there other modifications?

---

> ### Author Response · Authors · 2025-11-22
> **Response to Reviewer XiuL (Part 1)**
>
> Thank you for the valuable comments. Below, we have carefully addressed each of your comments.
>
> > **`W1`**: We would like to clarify that **graph learning  and KGE address fundamentally different learning problems**.
> >
> >**`W1.1`**: Graph learning methods operate on general graph-structured data where nodes are associated with feature vectors and labels, and the goal is typically node classification or graph-level prediction [1,2]. In contrast, KGE models encode entities and relations within triple-based structured knowledge, focusing on relational semantics rather than node features or label prediction [3]. Therefore, one-shot FKGE is **not a special case of** one-shot federated graph learning, **nor** can graph learning methods **be directly applied** to KGE due to the different data forms, learning objectives, and relational heterogeneity. We have added a short clarification in ***Appendix A.3 (line 814 - 823)*** to distinguish and compare them.
> >
> >**`W1.2`**: One-shot federated learning is merely a paradigm, and its core contribution lies in the processing, distillation, and aggregation of different data. Our work is **the first attempt to apply this paradigm to KGE**. In recent years, one-shot federated learning has indeed achieved remarkable development in various scenarios, but these methods [4,5] are mainly designed for image data and **cannot be directly transferred to KGE scenarios**.
>
> > **`W2`**: We have included relevant analysis in ***Appendix A.4.1 (line 873 - 898)***. The details are provided below.
> >
> >**`W2.1`**: The core idea of the one-shot federated learning approach we use is to **extract structured knowledge from each client’s local KGE model via knowledge distillation and transfer it to the global student model on the server in a single-shot communication**. The knowledge contained in a knowledge graph can be transmitted through the distillation process [6,8]. Specifically, assuming that the local KGs of all clients collectively form a global KG, each client can only observe a subset of the global KG, i.e., a local subgraph. Consequently, the client model captures the semantic and reasoning capabilities of this local subgraph, and the knowledge it encodes can be understood as the **semantic patterns between entities and relations** within the subgraph, which we refer to as ***knowledge***. On the server side, **knowledge distillation allows these local *knowledge* fragments from all subgraphs to be integrated into a global student model**. Based on this mechanism, extracting KG knowledge from client models via knowledge distillation and integrating it to form a global model is both reasonable and effective.
> >
> > **`W2.2`**: In our **one-shot** approach, the global model converges through **knowledge distillation on the server side**. Specifically, we first initialize the global model using an aggregation method such as FedAvg (other initialization strategies are detailed in ***Appendix A.5.3***), and then gradually acquire knowledge from client local models by optimizing the knowledge distillation objective.
> > $$
> > \mathcal{L} _ {distillation}= \alpha \mathcal{L} _ {kge} + (1-\alpha)\mathcal{L} _ {distill-soft}
> > $$
> > In each training step, operations such as gradient descent are applied to **minimize this objective on server**, allowing the student model to progressively approach the selected teacher models.
> >
> > In contrast, **multi-round** approaches rely on **iterative client-server interactions** to gradually achieve global convergence. Existing federated KGE methods [7,8] mainly perform FedAvg on the server side (e.g., ***Eqn 13***) and **minimize the KGE objective on clients** in each communication round.
> > $$
> > \mathcal{L} _ {kge} = \sum _ {(h,r,t) _ {\text{pos}}} \sum _ {(h', r, t') _ {\text{neg}}} \left[\gamma + f(h, r, t) - f(h', r, t')\right] _ +
> > $$
> > If multi-round methods are naively adapted to a one-shot setting, the server would only perform a single FedAvg, which is insufficient to fully transfer client knowledge.
> >
> > Therefore, our one-shot method addresses the issue that simple aggregation in multi-round methods cannot fully capture the diverse knowledge from clients in a one-shot scenario. After aggregation, knowledge distillation is applied to effectively transfer client knowledge to the global model, enabling it to **integrate more knowledge** than traditional aggregation methods.

---

> ### Author Response · Authors · 2025-11-22
> **Response to Reviewer XiuL (Part 2)**
>
> > **`W3`**: In this paper, 3 clients are selected by default in the experiments, mainly to clearly **demonstrate the effects and trends of OFKGE across different clients.** Additionally, referring to previous related studies (e.g., FedE [7], FedLU [8], pFedEG [9]), experiments typically tested up to 10 clients, so we adopted this setup. For example, **in Figure 3**, we actually extended the experiment to **10 clients** to verify the method’s stability and scalability in medium-scale scenarios.
> >
> > Regarding the large-scale scenarios you mentioned, we plan to evaluate the performance of ASA under larger client populations, specifically with 30, 50, and 100 clients. **These experiments are currently in progress** and are expected to be completed **within three days**. We will update both the manuscript and our responses to include the results once the experiments are finished.
>
> > **`W4`**: From both the perspectives of data quality and computational cost, **generating synthetic data is practical**. As shown in ***Figure 4***, we provide a visual illustration of the generated triples. In addition, in ***Appendix A.6.8 (line 1404 - 1434)***, we include a sensitivity analysis of the generated data, which shows that the data quality varies with hyperparameters. Therefore, in practical applications, using either publicly available unlabeled data or synthetic data is **feasible**：
> >
> > - If publicly **available unlabeled data** are accessible, they can be used directly [10].
> > - If such data are unavailable, **synthetic data** can be generated (see Appendix A.5.5) to capture global relation patterns and transfer teacher knowledge to the global student model. This strategy has established practice in the one-shot federated learning domain [4,11,12,13].
>
> > **`W5`**: When designing the teacher selection mechanism, we chose to operate at the relation level rather than the entity level, based on the following considerations:
> >
> > - In knowledge graphs, **relations are generally more important than individual entities** [14, 15]. A *relation* captures the semantic connections and patterns between entities, whereas a single *entity* lacks independent semantic significance.
> > - The goal of teacher selection is **to transfer valuable reasoning patterns**. Selecting at the relation level can capture the global interaction patterns between entities, making distillation more effective. In contrast, selecting teachers based on entities is difficult to reflect the structural and semantic rules in the graph.
>
> > **`Q1`**: Currently, there is no unified standard for quantitatively evaluating the quality of synthetic data [13,16]; it is mostly assessed indirectly by the improvement in the model’s final performance [16,17]. However, we observed that key hyperparameters in the generation process, such as the hyperparameter $\lambda$ (see Eqn 28 in ***Appendix A.5.5***), can affect the quality of the synthetic data, which in turn indirectly influences the distillation performance of OFKGE. The performance variations under **different $\lambda$ settings on the FB15K237 dataset** are shown in the table below, where the MRR and Hit@k metrics represent the average performance across all clients. The relevant modifications have been added to ***Appendix A.6.8 (line 1404 - 1434)***.
> >
> > | Hyperparameter | MRR       | Hit@10    | Hit@3     | Hit@1     |
> > | -------------- | --------- | --------- | --------- | --------- |
> > | $\lambda=0.0$  | 31.11     | 56.90     | 38.26     | 17.67     |
> > | $\lambda=0.2$  | **31.17** | **56.94** | **38.32** | **17.75** |
> > | $\lambda=0.4$  | 30.88     | 56.53     | 37.98     | 17.55     |
> > | $\lambda=0.6$  | 30.86     | 56.47     | 37.96     | 17.53     |
> > | $\lambda=0.8$  | 30.86     | 56.54     | 37.89     | 17.53     |
> > | $\lambda=1.0$  | 30.89     | 56.48     | 37.88     | 17.59     |

---

> ### Author Response · Authors · 2025-11-22
> **Response to Reviewer XiuL (Part 3)**
>
> > **`Q2`**: These baseline methods, such as FedE [7] and FedLU [8], were originally designed for multi-round FKGE. To ensure a fair comparison with our one-shot setting, we adapted them by **running only a single round while keeping all other settings unchanged**. In addition, we also compared their multi-round versions on the FB15K237 dataset (see the table below). The results show that their performance is comparable to our One-shot method, while our approach incurs significantly lower communication and computational costs, further validating the efficiency and effectiveness advantages of our approach.
> >
> > | Methods             | MRR   | Hit@10 | Hit@3 | Hit@1 |
> > | ------------------- | ----- | ------ | ----- | ----- |
> > | FedE (multi-round)  | 32.66 | 56.18  | 39.19 | 20.25 |
> > | FedLU (multi-round) | 33.71 | 57.53  | 40.21 | 21.27 |
> > | OFKGE (One-shot)    | 31.17 | 56.94  | 38.32 | 17.75 |
>
> We hope the above rebuttal resolves your concerns. If you still have questions, we are willing to answer your further questions.
>
> ***Reference***:
>
> > [1] Wan, G., Huang, W., & Ye, M. (2024). Federated graph learning under domain shift with generalizable prototypes. *AAAI*.
> >
> > [2] Qian, J., Wan, G., Huang, W., et al. (2025). GHOST: Generalizable One-Shot Federated Graph Learning with Proxy-Based Topology Knowledge Retention. *ICML*.
> >
> > [3] Cao, J., Fang, J., Meng, Z., & Liang, S. (2024). Knowledge graph embedding: A survey from the perspective of representation spaces. *ACM Computing Surveys*.
> >
> > [4] Bai, J., Song, Y., Wu, D., et al. (2025). A unified solution to diverse heterogeneities in one-shot federated learning. *KDD*.
> >
> > [5] Chen, H., Li, H., Zhang, Y., et al. (2025). Fedbip: Heterogeneous one-shot federated learning with personalized latent diffusion models. *CVPR*.
> >
> > [6] Liu, J., Wang, P., Shang, Z., & Wu, C. (2023). Iterde: an iterative knowledge distillation framework for knowledge graph embeddings. *AAAI*.
> >
> > [7] Chen, M., Zhang, W., Yuan, Z., et al. (2021). Fede: Embedding knowledge graphs in federated setting. *IJCKG*.
> >
> > [8] Zhu, X., Li, G., & Hu, W. (2023). Heterogeneous federated knowledge graph embedding learning and unlearning. *WWW*.
> >
> > [9] Zhang, X., Zeng, Z., Zhou, X., et al. (2025). Personalized federated knowledge graph embedding with client-wise relation graph. *Applied Intelligence*.
> >
> > [10] Li, Q., He, B., & Song, D. (2021). Practical One-Shot Federated Learning for Cross-Silo Setting. *IJCAI*.
> >
> > [11] Zhang, J., Chen, C., Li, B., et al. (2022). Dense: Data-free one-shot federated learning. *NeurIPS*.
> >
> > [12] Heinbaugh, C. E., Luz-Ricca, E., & Shao, H. (2023). Data-free one-shot federated learning under very high statistical heterogeneity. *ICLR*.
> >
> > [13] Dai, R., Zhang, Y., Li, A., et al. (2024). Enhancing One-Shot Federated Learning Through Data and Ensemble Co-Boosting. *ICLR*.
> >
> > [14] Ji, S., Pan, S., Cambria, E., et al. (2021). A survey on knowledge graphs: Representation, acquisition, and applications. *IEEE transactions on neural networks and learning systems*.
> >
> > [15] Yue, L., Zhang, Y., Yao, Q., et al. (2023). Relation-aware Ensemble Learning for Knowledge Graph Embedding. *EMNLP*.
> >
> > [16] Zhao, S., Liao, T., Fu, L., Chen, C., Bian, J., & Zheng, Z. (2024). Data-free knowledge distillation via generator-free data generation for Non-IID federated learning. *Neural Networks*.
> >
> > [17] Zeng, H., Xu, M., Zhou, T., Wu, X., Kang, J., Cai, Z., & Niyato, D. (2024, October). One-shot-but-not-degraded Federated Learning. *MM*.

---

> ### Author Response · Authors · 2025-11-25
> **Supplementary Answer to Reviewer XiuL (Weakness 3)**
>
> >**`W3 (Supplementary)`**:
> >
> > We have completed the additional experiments under larger client scales (30, 50, 80, and 100 clients). The results are summarized in the table below. Across all settings, OFKGE consistently outperforms the Single method, demonstrating that ASA remains effective even as the number of clients substantially increases. Performance gains are observed in MRR, Hit@3, and Hit@1, while Hit@10 remains comparable or slightly improved. These results further **validate that OFKGE can scale to real large-scale scenarios**. We have included these results in ***Appendix A.6.9 (line 1435 - 1457)*** of the revised manuscript.
> >
> > | #Clients | Method |    MRR    |  Hit@10   |   Hit@3   |   Hit@1   |
> > | :------: | :----: | :-------: | :-------: | :-------: | :-------: |
> > |    30    | Single |   29.23   |   53.03   |   35.34   |   16.97   |
> > |          | OFKGE  | **30.06** | **53.95** | **36.15** | **17.78** |
> > |    50    | Single |   29.58   |   52.64   |   35.55   |   17.64   |
> > |          | OFKGE  | **30.44** | **53.54** | **36.40** | **18.50** |
> > |    80    | Single |   30.01   |   52.40   |   36.01   |   18.28   |
> > |          | OFKGE  | **30.93** | **53.27** | **36.84** | **19.28** |
> > |   100    | Single |   30.31   |   52.35   |   36.30   |   18.71   |
> > |          | OFKGE  | **31.23** | **53.20** | **37.12** | **19.74** |

---

> > ### Comment · Reviewer_XiuL · 2025-11-27
> >
> > Thanks a lot for the reviewers' rebuttal. For weakness 1, I would like to clarify that the one-shot idea proposed in the referenced paper I gave can indeed be adapted to knowledge graph embedding. KGE and general graph learning share a common foundation by learning node representations. So the underlying strategy naturally transfers.
> >
> > After reading the comments proposed by other reviewers, i would like to remain my score.

---

> > > ### Author Response · Authors · 2025-11-29
> > >
> > > We would like to emphasize that [1] is currently only an **arXiv preprint** and has not undergone peer review. Its methods and conclusions have not been fully validated. In the absence of empirical evidence, inferring that its strategy can “*naturally transfer*” to KGE solely because both GL and KGE involve node representations **is not rigorous**. On one hand, [1] does not discuss or evaluate KGE scenarios, so its applicability to KGE cannot be assumed; on the other hand, drawing a simple analogy between GL and KGE ignores the fundamental differences between the tasks.
> > >
> > >
> > > **`Fundamental differences`**: Indeed, as you pointed out, *“KGE and general graph learning share a common foundation by learning node representations.”* We agree that, conceptually, both KGE and GL involve node (entity) representation learning, and the one-shot idea is conceptually inspiring. However, this does not imply that adapting [1] to KGE is as simple as *replacing nodes with entities*. We would like to clarify that **KGE and GL differ fundamentally in both data structure and learning objectives.**
> > >
> > > - **Data structure:** `[1]’s graphs` are undirected, with nodes carrying features and labels, and edges representing only topological connections for neighborhood aggregation, without semantic meaning. In contrast, `KG data` consist of structured triples $(h, r, t)$ with diverse entity and relation types [2]. Relations carry semantics, connect entities, and constrain triple validity . Therefore, [1] **cannot capture the multi-relational semantics of KGs.**
> > > - **Learning objectives:** KGE emphasizes triple-level semantic plausibility, while [1] focuses on node distinguishability.`[1] primarily targets` node classification or graph-level prediction, learning node representations that differentiate labels via neighbor aggregation. `KGE` focuses on semantic reasoning (e.g., link prediction or triple scoring), learning entity and relation representations that model the plausibility of triples [2]. Consequently, [1] optimizes node representations under topological constraints, whereas **KGE emphasizes semantic modeling of triples while jointly optimizing entity and relation representations**, highlighting a fundamental difference in objectives.
> > >
> > >
> > > **`Technical Challenges`**: After reviewing the paper, we also attempted to adapt [1] to KGE but **found several technical challenges**:
> > >
> > > - **Client-side node statistics cannot capture triple semantics:** [1] uploads statistics based on node feature distributions, which cannot reflect the semantic constraints of *entity–relation–entity.*
> > > - **Server-side aggregation is incompatible:** [1] first aggregates node feature distributions to obtain global mean and variance, then initializes a small set of nodes and optimizes both node features and edge existence, producing a *topologically consistent, feature-aligned* global surrogate graph. However, this approach only generates global node representations without learning or aggregating global relation representations. Moreover, the surrogate graph is limited (≤300 nodes), insufficient to capture the diversity of entities and relations in a KG, and cannot summarize global semantics. Even if generated, it may capture structural patterns but lacks semantic constraints, potentially leading to semantically inconsistent graphs.
> > > - **Client-side fine-tuning based on surrogate graphs cannot be directly applied:** [1]’s distillation objective is class probabilities, whereas KGE distillation should focus on triple plausibility scores. Furthermore, [1]’s fine-tuning adapts only node features, while KGE fine-tuning must account for triple semantics and structural constraints.
> > >
> > > Therefore, adapting [1] to KGE would require redefining client-uploaded information, redesigning server-side aggregation, and developing a new client-side distillation mechanism. **This goes far beyond “*naturally transferring*” the method. It amounts to redesigning a one-shot federated framework specifically for KGE.** Our OFKGE framework was explicitly designed to address these specific challenges. For example, the ASA mechanism dynamically selects the best teacher to achieve fine-grained semantic transfer, and client-side regularized fine-tuning preserves local triple characteristics while integrating global semantics. We thus believe that our work represents the first one-shot federated framework tailored for KGE, whereas [1] cannot be directly applied to the KGE scenario.
> > >
> > > ***Reference***:
> > >
> > > > [1] Yan, G., Li, X., Xie, L., Zhang, W., Shen, Q., Fang, Y., & Wu, Z. (2025). Personalized One-shot Federated Graph Learning for Heterogeneous Clients. *arXiv preprint arXiv:2411.11304*.
> > > >
> > > > [2] Cao, J., Fang, J., Meng, Z., & Liang, S. (2024). Knowledge graph embedding: A survey from the perspective of representation spaces. *ACM Computing Surveys*.

---

### Official Review · Reviewer_jcKT · 2025-10-31

**Soundness:** 3
**Presentation:** 4
**Contribution:** 2
**Rating:** 6
**Confidence:** 3

**Summary:**

This paper addresses critical limitations of existing Federated Knowledge Graph Embedding methods. It proposes One-Shot Federated Knowledge Graph Embedding, a framework built on the One-Shot Federated Learning paradigm, which streamlines training into three core stages:

i) One-Shot Model Distillation: Each client independently trains a local Knowledge Graph Embedding model using its private knowledge graph and uploads the model parameters and relation-aware performance weights to the server only once, eliminating iterative communication.

ii) Server-Side Adaptive Knowledge Integration: The server leverages an Adaptive Semantic Alignment mechanism to dynamically select the most informative client model as a teacher for each batch of distillation data. It then performs knowledge distillation to fuse multi-source semantics into a generalized global embedding, which is distributed back to clients as auxiliary knowledge.

iii) Client-Side Local Fine-Tuning: Clients refine their local models using the global auxiliary knowledge and a regularization-based strategy, preserving personalized local KG structures while enhancing inference performance with global semantics.

**Strengths:**

In innovation aspect, OFKGE is the first work to adapt OFL to FKGE, addressing the root cause of high communication costs in multi-round FKGE methods. This innovation fills a gap in applying OFL to knowledge graph tasks, where structural and semantic complexity differs from computer visio.

In experimental aspect, covers multimodal (MKG-W, MKG-Y) and standard (FB15K-237) KGs to test adaptability across data types. Includes ablation studies (validating ASA, aggregation initialization, and fine-tuning), cost analysis, hyperparameter sensitivity tests, and low-overlap scenario evaluations, providing rigorous evidence for OFKGE’s effectiveness.

In application aspect, OFKGE is well-suited for resource-constrained (low bandwidth) and privacy-sensitive scenarios (e.g., healthcare/finance KGs) and supports future extension to multimodal KGs, offering clear real-world value .

**Weaknesses:**

i) The baseline methods used for comparison are somewhat outdated and fail to cite recent approaches such as "Privacy-Preserving Federated Embedding Learning for Localized Retrieval-Augmented Generation."

ii) Line 466, the paper states, "Despite this semantic gap and the presence of some factual errors, training with synthetic triples still enhances the distillation process." Intuitively, data containing grammatical and factual errors would negatively impact model distillation, particularly for embedding models. However, the paper presents a completely opposite viewpoint without providing explanations or references.

iii) The experimental analysis section is relatively weak, primarily emphasizing that the author's model achieved strong results in evaluations, but lacking in-depth analysis of the experimental findings and insightful conclusions that could guide the field forward.

**Questions:**

As I mentioned in the weaknesses section:

Could you add citations, comparisons, and analyses of some recent methods, such as the method "Privacy-Preserving Federated Embedding Learning for Localized Retrieval-Augmented Generation."?

I have a question regarding “Despite this semantic gap and the presence of some factual errors, training with synthetic triples still enhances the distillation process”—could you provide a further explanation?

---

> ### Author Response · Authors · 2025-11-22
> **Response to Reviewer jcKT**
>
> Thank you for the valuable comments. Below, we have carefully addressed each of your comments.
>
> > `[W1 & Q1]`:  We have reviewed this work; however, it is currently available only as an arXiv preprint [1] and has not yet undergone peer review, so we have not included it in our quantitative comparisons. The baseline methods we selected, such as FedLU [2], FedR [3], and the recent work pFedEG [4], have all been peer-reviewed, ensuring that the comparisons are both timely and relevant. Moving forward, we will continue to keep track of the progress of this work. Meanwhile, we have added the citation to this work in our paper and included the analysis in the ***Detailed Related Works*** section (***Appendix A.3***, **line 814 - 823**).
>
> > `[W2 & Q2]`:  Regarding the explanation of the effectiveness of using synthetic triples, we have revised ***Section 5.8 (line 469 - 470)*** and added supplementary details in ***Appendix A.6.8 (line 1406 - 1434)***. We further clarify as follows:
> >
> > - During the knowledge distillation process, we train using a combination of soft and hard labels rather than relying solely on the absolute correctness of each triple. This means that **the distillation learns overall relational patterns** rather than fitting every single triple exactly. Therefore, moderate semantic deviations or occasional factual errors have limited impact on the global knowledge. Moreover, this conclusion **is consistent with existing studies on knowledge distillation using synthetic data**, such as [5, 6, 7, 8]. Hence, synthetic data is both feasible and effective for facilitating knowledge distillation.
> > - **Experimental results** (*Table 1*) demonstrate that distilling global knowledge with synthetic data helps local models achieve performance improvements, outperforming other baseline methods. This indicates that synthetic data indeed plays a positive role in server-side knowledge distillation.
>
> > `[W3]`:  We have made corresponding revisions in ***Section 5.4 (line 319 - 323) and Appendix A.6.6 (line 1360 - 1363)***, conducting a more in-depth analysis of the reasons for the factors underlying the observed performance changes in the experimental results. For a detailed overview of these modifications, please refer to the revised manuscript.
>
> We hope the above rebuttal resolves your concerns. If you still have questions, we are willing to answer your further questions.
>
> ***Reference***:
> > [1] Mao, Q., Zhang, Q., Hao, H., Han, Z., Xu, R., Jiang, W., ... & Yu, P. S. (2025). Privacy-preserving federated embedding learning for localized retrieval-augmented generation. *arXiv preprint arXiv:2504.19101*.
> >
> > [2] Zhu, X., Li, G., & Hu, W. (2023, April). Heterogeneous federated knowledge graph embedding learning and unlearning. In *Proceedings of the ACM web conference 2023* (pp. 2444-2454).
> >
> > [3] Zhang, K., Wang, Y., Wang, H., Huang, L., Yang, C., Chen, X., & Sun, L. (2022, December). Efficient Federated Learning on Knowledge Graphs via Privacy-preserving Relation Embedding Aggregation. In *Findings of the Association for Computational Linguistics: EMNLP 2022* (pp. 613-621).
> >
> > [4] Zhang, X., Zeng, Z., Zhou, X., Niyato, D., & Shen, Z. (2025). Personalized federated knowledge graph embedding with client-wise relation graph. *Applied Intelligence*, *55*(5), 318.
> >
> > [5] Zhu, Z., Hong, J., & Zhou, J. (2021, July). Data-free knowledge distillation for heterogeneous federated learning. In *International conference on machine learning* (pp. 12878-12889). PMLR.
> >
> > [6] Zhang, J., Chen, C., Zhuang, W., & Lyu, L. (2023). Target: Federated class-continual learning via exemplar-free distillation. In *Proceedings of the IEEE/CVF International Conference on Computer Vision* (pp. 4782-4793).
> >
> > [7] Dai, R., Zhang, Y., Li, A., Liu, T., Yang, X., & Han, B. (2024). Enhancing One-Shot Federated Learning Through Data and Ensemble Co-Boosting. In *The Twelfth International Conference on Learning Representations*.
> >
> > [8] Zeng, H., Xu, M., Zhou, T., Wu, X., Kang, J., Cai, Z., & Niyato, D. (2024, October). One-shot-but-not-degraded Federated Learning. In *Proceedings of the 32nd ACM International Conference on Multimedia* (pp. 11070-11079).

---

### Official Review · Reviewer_jYhd · 2025-11-01

**Soundness:** 3
**Presentation:** 2
**Contribution:** 2
**Rating:** 6
**Confidence:** 3

**Summary:**

This manuscript introduces OFKGE, an efficient One-Shot Federated Knowledge Graph Embedding framework that tackles the high communication and privacy costs of traditional iterative methods. Clients train local knowledge graph models and upload them a single time; the server then intelligently distills a global knowledge representation using a novel Adaptive Semantic Alignment (ASA) mechanism that dynamically selects the best "teacher" model from the clients. This distilled global model is then sent back to clients, who fine-tune their local embeddings with it, balancing generalized knowledge with local personalization.

**Strengths:**

- The paper's primary strength is its one-shot approach. By eliminating iterative communication rounds, it drastically reduces the communication overhead and shrinks the attack surface for privacy violations like gradient inversion attacks. This makes the framework highly practical for real-world scenarios with limited bandwidth or strict privacy requirements.
- The server-side is a sophisticated alternative to naive model averaging. Instead of simply blending all client models, it dynamically selects the most informative client model (the "best teacher") for specific relations during distillation. This allows the server to build a more nuanced and accurate global model by drawing on the specialized strengths of different clients, effectively mitigating the negative impact of data heterogeneity.
- The framework concludes with a client-side fine-tuning stage that uses a dedicated regularization term to balance two objectives: integrating the powerful distilled global knowledge and preserving the unique, personalized characteristics of the client's original local model.

**Weaknesses:**

- The effectiveness of the server-side distillation, particularly the ASA "best teacher" selection, is highly dependent on the quality of the uploaded client models and their self-reported performance metrics. If a client's model is poorly trained, biased, or overfitted to its local data, it can be selected as a teacher and subsequently degrade the quality of the global model for all other clients. The framework assumes all clients provide high-quality, reliable models, which may not hold true in practice.
- The one-shot nature of the framework is also its main weakness. Real-world knowledge graphs are dynamic and constantly evolving with new entities and relations. OFKGE provides no mechanism for clients to incorporate new knowledge or for the global model to be updated after the single communication round. This static "snapshot" approach makes it unsuitable for long-term deployments or real-time systems where continuous learning is essential.

**Questions:**

- How could the OFKGE framework be extended to handle dynamic KGs where new entities, relations, or triples are added over time? Would this require periodic re-uploads from clients, a mechanism for incremental updates to the global model, or an entirely different approach that moves beyond the one-shot paradigm?
- In a massively federated scenario with hundreds or thousands of clients, how do you expect the ASA mechanism to perform? Would the process of selecting a single "best teacher" per batch remain effective, or would a more complex aggregation of multiple top-performing teachers be necessary to capture the diverse knowledge present across a large client population?

---

> ### Author Response · Authors · 2025-11-22
> **Response to Reviewer jYhd**
>
> Thank you for the valuable comments. Below, we have carefully addressed each of your comments.
>
> > `[W1]`:
> >
> > In OFKGE, the core goal of the ASA *best teacher* selection is to **transfer the most valuable relation-level knowledge**. The selection mechanism relies on **relation-level performance metrics** and dynamically chooses teacher models for each batch. Specifically, client models are first screened based on the entities involved in the batch to form a candidate set; then, using the relation-level performance matrix, the client that performs best on the relations present in the batch is selected as the optimal teacher.
> >
> > Importantly, **this mechanism does not require all clients to provide high-quality models**. The relation-level performance matrix naturally captures each client’s relative competence across different relations. For example, a client may perform well on relation A (with abundant data) but poorly on relation B (with sparse data). In such cases, the client will only be selected as a teacher for batches involving its strong relations, and will rarely influence batches involving weaker relations. Low-quality models, which typically perform poorly across most relations, are thus seldom chosen as the best teacher.
> >
> > As a result, **even when some client models are suboptimal, the best teacher selection mechanism remains effective**, ensuring that the global distillation process is stable and reliable.
>
>
> > `[W2 & Q1]`:
> >
> > (a) Using the *one-shot paradigm* is not a weakness but an efficient and flexible training strategy that is **well suited for long-term deployment and dynamic update scenarios**. Importantly, the *one-shot paradigm* does not mean the global model is trained only once forever, but that each update event only requires one aggregation round. As a result, OFKGE can be **executed repeatedly whenever new entities, relations, or triples appear**, allowing the global model to incorporate the latest client-side knowledge with minimal communication and computation.
> >
> > (b) Upon the arrival of new data, a single OFKGE run is sufficient to update the global model. There is no need for clients to periodically re-upload their data, nor is it necessary to design complex incremental update mechanisms. Compared with multi-round federated learning, the one-shot paradigm avoids continuous communication and the maintenance cost of incremental updates. **Thus, a single execution triggered by new data is enough to refresh global knowledge**, making the approach naturally suitable for dynamic KG environments.
>
> > `[Q2]`:
> >
> > (a) In large-scale federated scenarios, the ASA mechanism **remains effective**, which we attribute primarily to the **the per-batch dynamic selection at relation-level granularity**. Even with a large number of clients, the candidate set remains limited after an initial screening that checks whether the entities involved in the batch exist on each client. The *best teacher* for each batch is then selected only from the candidates that perform best on the relations involved. This dynamic and localized selection ensures that each batch receives guidance from the most relevant clients, without being diluted by the sheer number of clients.
> >
> > (b) Moreover, the ASA mechanism can **flexibly be extended to aggregate multiple top-performing teachers per batch**. For example, the top-k clients can be combined with weighted contributions based on relation-level performance metrics, providing richer knowledge while still focusing on the most competent clients for each relation.  We plan to evaluate the performance of ASA under larger client populations, specifically with 30, 50, and 100 clients. **These experiments are currently in progress** and are expected to be completed **within three days**. We will update both the manuscript and our responses to include the results once the experiments are finished.
>
> We hope the above rebuttal resolves your concerns. If you still have questions, we are willing to answer your further questions.

---

> ### Author Response · Authors · 2025-11-25
> **Supplementary Answer to Reviewer jYhd (Question 2)**
>
> >**`Q2 (Supplementary)`**:
> >
> > We have conducted additional experiments under larger client scales (30, 50, 80, and 100 clients), and the results are summarized in the table below. Across all settings, OFKGE consistently outperforms the Single method, demonstrating that the ASA mechanism, using a single best-teacher per batch, **remains effective** even as the number of clients substantially increases. These results have been added to ***Appendix A.6.9 (lines 1435–1457)*** of the revised manuscript.
> >
> > | #Clients | Method |    MRR    |  Hit@10   |   Hit@3   |   Hit@1   |
> > | :------: | :----: | :-------: | :-------: | :-------: | :-------: |
> > |    30    | Single |   29.23   |   53.03   |   35.34   |   16.97   |
> > |          | OFKGE  | **30.06** | **53.95** | **36.15** | **17.78** |
> > |    50    | Single |   29.58   |   52.64   |   35.55   |   17.64   |
> > |          | OFKGE  | **30.44** | **53.54** | **36.40** | **18.50** |
> > |    80    | Single |   30.01   |   52.40   |   36.01   |   18.28   |
> > |          | OFKGE  | **30.93** | **53.27** | **36.84** | **19.28** |
> > |   100    | Single |   30.31   |   52.35   |   36.30   |   18.71   |
> > |          | OFKGE  | **31.23** | **53.20** | **37.12** | **19.74** |

---

### Author Response · Authors · 2025-12-03
**Summary of Rebuttal**

Dear PCs, Senior ACs, and ACs,

We sincerely thank reviewers `jYhd`, `jcKT`, `XiuL`, and `jtAW` for their time and constructive feedback. As the discussion period concludes, we would like to summarise our rebuttal and the exchanges that took place.

1. **Dynamic Adaptability of One-Shot Paradigm**（`jYhd`）
   - In response to `jYhd`’s concern that the one-shot paradigm may not suit dynamic environments, we clarify that one-shot FL is naturally compatible with dynamic KGs. When new data arrives, OFKGE can simply be re-run once without requiring additional mechanisms or multi-round communication.

2. **Additional Experiments and Clarifications**（`jYhd`、`XiuL`、`jcKT`）

   - Regarding the scalability concerns raised by `jYhd` and `XiuL`, we **conducted experiments** on FB15k-237 with 30 / 50 / 80 / 100 clients (***Appendix A.6.9, line 1435 - 1457*** ), showing ASA remains effective as client count increases.

   - For the baseline selection concerns raised by `jcKT`, we clarified our choices and added further references and comparisons in ***Appendix A.3 (line 814–823)***. To strengthen the experimental analysis that `jcKT` found thin, we revised ***Section 5.4 (line 319–323) and Appendix A.6.6 (line 1360–1363)*** to include a deeper analysis of the factors behind observed performance differences.

   - Concerning `XiuL`’s question about adapting multi-round baselines to the one-shot setting, we explained our approach and **compared the performance of multi-round FedE / FedLU with OFKGE** on FB15k-237. The results show a small performance gap, while OFKGE substantially reduces communication and computation costs.

3. **Synthetic Data**（`jcKT`、`XiuL`）

   - Regarding `jcKT`’s confusion about synthetic data (Line 466), we provided a detailed explanation supported by both literature references and empirical evidence. These clarifications appear in ***Section 5.8 (lines 469–470) and Appendix A.6.8 (lines 1406–1434)***.

   - Addressing `XiuL`’s concern on the practicality and sensitivity of synthetic data, we present ***Figure 4*** and add sensitivity experiments on key generation hyperparameters (***Appendix A.6.8, lines 1404–1434***), which show how data quality varies with hyperparameters. We further clarified, with references, that using publicly available unlabeled or synthetic data is a  feasible practice.

4. **Effectiveness of ASA**（`jYhd`、`XiuL`、`jtAW`）

   - Regarding `jYhd`’s concern about low-quality clients, we expanded the description of ASA’s relation-aware teacher selection workflow and illustrated why the mechanism remains effective.

   - To address `XiuL` and `jtAW`’s question about using relation-aware performance matrices, we explained that relations are the primary semantic connectors in KGs, often more informative than individual entities, making selection more effective for transferring reasoning patterns. We further supported this claim with references to existing studies.

5. **Clarifications on Novelty**（`XiuL`、`jtAW`）

   - Regarding `XiuL`’s suggestion that existing one-shot GL methods can be directly transferred to KGE, we note that **this reflects a misunderstanding**, as it equates two tasks that are fundamentally different. We clarified the fundamental differences between GL and KGE and detailed technical challenges in applying the referenced work to KGE, which require substantial redesign of core components. We also added theoretical analysis of OFL’s effectiveness on KGE in **Appendix A.4.1 (lines 873–898)**.

   - Regarding `jtAW`’s comment that adapting OFL to KGE is merely a simple combination of existing methods, we pointed out that **this overlooks the unique challenges posed by KG data**. We summarized prior research showing that FL and OFL face distinct challenges across different data modalities, emphasizing that these solutions are nontrivial. We further explained the additional challenges specific to KGs. Concerning `jtAW`’s privacy concerns, we clarified that this work focuses on framework design rather than privacy guarantees, leaving formal enhancements to future work.

We sincerely thank reviewers `jYhd` and `jcKT` for **recognizing the value and contributions of our work**. We greatly appreciate their support. Regarding reviewers `XiuL` and `jtAW`, we note that there appear to be some misunderstandings and biases that we would like to clarify. Reviewer `XiuL` **conflates GL and KGE**, but the two differ fundamentally in data structure, method design, and objectives, as we have explained in our rebuttal. Reviewer `jtAW` seems to **overlook unique challenges posed by KG data**, which cannot be addressed through simple transfer. In our rebuttal, we provided examples of research on FL and OFL across various data modalities, highlighting that designing solutions to domain-specific challenges is valuable. Since `jtAW` does not fully consider the examples and explanations we provided, we feel that this assessment may not fairly represent our contributions.

---

### Meta-Review · Area_Chair_AEid · 2026-01-06

**Summary:**

The paper proposes OFKGE, a one-shot federated learning framework for knowledge graph embedding (KGE). The pipeline has three stages: (1) client-side local training of KGE models, uploaded once with relation-aware performance weights; (2) server-side distillation using an Adaptive Semantic Alignment (ASA) mechanism that selects the strongest client “teacher” per batch to fuse heterogeneous knowledge into a global embedding; and (3) client-side fine-tuning with a regularizer to balance global semantics and local personalization. Experiments of multiple KGs indicate improved efficiency and competitive accuracy.

hough some concerns were resolved, some key concerns were not well addressed as listed in the "Reviewer Concerns" section.

**Reviewer Concerns:**

The concerns like bad client may harm overall performance (jYhd, XiuL), OFKGE provides no mechanism for dynamic KGs (jYhd), synthetic data may not always be available or practical (XiuL) were resolved by the rebuttal. The following reviewer concerns were not addressed and some are still outstanding:

1/ Noisy synthetic triples with grammatical or factual errors still enhances the distillation process is counterintuitive. (jcKT) The rebuttal lacks a concrete analysis that quantifies the noise level at which synthetic triples help vs. hurt performance.

2/ The experimental analysis section lacks in-depth analysis of the experimental findings and insightful conclusions. (jcKT)

3/ This is no convergence analysis or theoretical bounds on the performance gap between one-shot and multi-round approaches. (XiuL)

4/ Discussions of some related works are missing in the main paper. (jcKT, XiuL)

5/ Experiments are conducted with small number of clients, which limits the working scenario of federated learning (XiuL). Only one experiments were conducted with 10 clients as in Figure 3. In the rebuttal, the authors did not extend the experiments to more datasets.

6/ The multi-teacher distillation method in Stage 2 is not novel. (jtAW)

7/ Empirical evidence of privacy protection capabilities of OFKGE is missing. (jtAW)

**Reviewer Scores:**

Two reviewers gave positive scores (two 6s) and two reviewers gave negative scores (2 and 4). I do not think reviewers would change their scores.

---

### Decision · Program_Chairs · 2026-01-26

Reject